# Layer-specific chromatin accessibility landscapes reveal regulatory networks in adult mouse visual cortex

**Lucas T Gray, Zizhen Yao, Thuc Nghi Nguyen, Tae Kyung Kim, Hongkui Zeng, Bosiljka Tasic***

Allen Institute for Brain Science, Seattle, United States

**Abstract** Mammalian cortex is a laminar structure, with each layer composed of a characteristic set of cell types with different morphological, electrophysiological, and connectional properties. Here, we define chromatin accessibility landscapes of major, layer-specific excitatory classes of neurons, and compare them to each other and to inhibitory cortical neurons using the Assay for Transposase-Accessible Chromatin with high-throughput sequencing (ATAC-seq). We identify a large number of layer-specific accessible sites, and significant association with genes that are expressed in specific cortical layers. Integration of these data with layer-specific transcriptomic profiles and transcription factor binding motifs enabled us to construct a regulatory network revealing potential key layer-specific regulators, including *Cux1/2*, *Foxp2*, *Nfia*, *Pou3f2*, and *Rorb*. This dataset is a valuable resource for identifying candidate layer-specific cis-regulatory elements in adult mouse cortex.

## Introduction

*For correspondence: bosiljkat@alleninstitute.org

**Competing interests:** The authors declare that no competing interests exist.

Many complex functions carried out by mammalian brains arise through the concerted efforts of different cell types in the neocortex. The cortex is organized during development into a laminar structure, with each layer composed of a distinct set of cell types with different morphological, electrophysiological, and connectional properties. These diverse cellular phenotypes are established and maintained by complex interactions of sequence-specific transcription factors (TFs), and they are reinforced by chromatin modifiers. Defining cell-type specific chromatin signatures will provide an understanding of the regulatory landscapes that influence transcription, as well as identification of putative cell type-specific cis-regulatory elements. Most variation associated with phenotypic differences in humans identified through genome-wide association studies is located in non-coding regions of the genome (*Albert and Kruglyak, 2015*; *Tak and Farnham, 2015*). However, the exact function of these polymorphisms and the cell types that they affect is established only in a minority of cases (*Soldner et al., 2016*). Assigning regulatory elements to cell types provides insight into their potential function (*Corces et al., 2016*). In addition, specific regulatory elements could be used to build cell-type specific genetic tools similar to those created by careful selection of well-studied regulatory regions near genes (*Bou-Gharios et al., 1996*; *Pinkert et al., 1987*) or by large genomic screens (*Shima et al., 2016*).

Until recently, two major hurdles have restricted access to epigenetic landscapes of specific primary cell types: selective access to those types, and the large numbers of cells required as an input for epigenomic characterization using ChIP-seq, DNAse-seq, or FAIRE-seq. Recent development of a number of Cre-recombinase transgenic lines allowed access to specific cortical cell types (*Harris et al., 2014*), and has enabled transcriptomic characterization and classification of cells from the visual cortex using single-cell RNA-seq (scRNA-seq) (*Tasic et al., 2016*). Though these transgenic

lines have differing degrees of cell type heterogeneity, they provide a platform for accessing populations of related cell types from different layers of the cortex. New techniques, including ATAC-seq (*Buenrostro et al., 2013*) and THS-seq (*Sos et al., 2016*), enable measurement of chromatin accessibility from a few hundred or even single cells (*Buenrostro et al., 2015*; *Corces et al., 2016*; *Cusanovich et al., 2015*; *Lara-Astiaso et al., 2014*).

Recent studies have started to probe the correspondence between transcription and epigenetics in cortical cell types labelled by the expression of somatostatin (*Sst*), parvalbumin (*Pvalb*), and calcium/calmodulin-dependent protein kinase II alpha (*Camk2a*) from whole mouse cortex (*Mo et al., 2015*), rod and cone photoreceptor cells in retina (*Mo et al., 2016*), and in the developing and adult cerebellum (*Frank et al., 2015*). These studies have profiled the chromatin accessibility landscapes of broad cell classes, but did not examine layer-specific differences in the neocortex. Here, we take advantage of pan-GABAergic and layer-specific glutamatergic Cre-driver lines, low-input ATAC-seq, and fluorescence-activated cell sorting (FACS) to investigate chromatin accessibility landscapes in a specific cortical region, the adult mouse visual cortex (VISp).

Each layer of VISp contains distinct populations of glutamatergic cells with different transcriptional, functional, and connectional properties: layer 4 cells are the primary recipients of the visual signals from the dorsal portion of the lateral geniculate nucleus (LGd); layer 2/3 cells receive signals from L4, and communicate with L5 cells within the same cortical region and other cortical regions; layer 5 cells are highly diverse, and include cells that project to many other cortical and subcortical regions; and the layer 6 cells we examine in this study project to the thalamus (*Bortone et al., 2014*; *Sorensen et al., 2015*). Thus, within even a small region of the cortex, there is great diversity of cell types, each of which carries out a distinct transcriptional program (*Tasic et al., 2016*). However, the regulatory programs that produce these different transcriptional and cellular phenotypes are not known. In order to define potential regulatory elements and corresponding transcriptional regulators, we examined chromatin landscapes of these cell classes.

We found broad differences between GABAergic and glutamatergic cell types, as well as layer-specific differential chromatin accessibility in glutamatergic cell types that correlated with differential gene expression. Putative regulatory elements were identified through TF motif searches and comparisons to existing ChIP-seq datasets for each cell class. With these components, we built a putative regulatory network of TF binding sites near layer-specific TF genes that may govern layer-specific transcriptomic states. This network suggests that *Cux1/2, Foxp2, Nfia, Pou3f2, and Rorb* are key regulators for the maintenance of molecular identity of deep layer and upper-layer cortical cells.

## Results

### Layer-specific chromatin accessibility profiling by ATAC-seq

To access layer-specific glutamatergic cells in the mouse visual cortex, we used four previously characterized Cre lines crossed to the *Ai14* reporter line (*Madisen et al., 2010*), which expresses tdTomato (tdT) after Cre-mediated recombination (*Figure 1A,B*). Although these lines mostly label cells in specific cortical layers, we note that each contains at least two closely related cell types based on scRNA-seq (*Figure 1C*, *Tasic et al., 2016*). As a control, we profiled GABAergic cell types using *Gad2-IRES-Cre*. Because each of these Cre line-derived populations contains more than one transcriptomic cell type (*Tasic et al., 2016*), we will refer to these populations as cell classes. We tried to minimize other potential sources of heterogeneity that may be caused by age, sex, or cortical region by restricting our analysis to eight week-old male mice, and cells microdissected only from the visual cortex (*Figure 1A*). After protease treatment and trituration, cells were isolated by FACS (Materials and methods). We collected triplicates of 500 cell populations from each Cre line and from at least two mice per line.

The low-input assay for transposase-accessible chromatin (ATAC) was adapted from a previous study (*Lara-Astiaso et al., 2014*) (Materials and methods). As a control for the ATAC-seq assay, we profiled chromatin accesibility landscapes of 500-cell populations of mouse ES (mES) cells. Low-depth sequencing was performed to identify libraries that have high read diversity within mouse genome-aligned reads, indicating that the library did not consist of many PCR duplicates, as well as a characteristic fragment size pattern that demonstrates protection of DNA by nucleosomes. High-quality libraries were then sequenced using Illumina HiSeq or MiSeq (min: 13.2 M, median: 83 M,

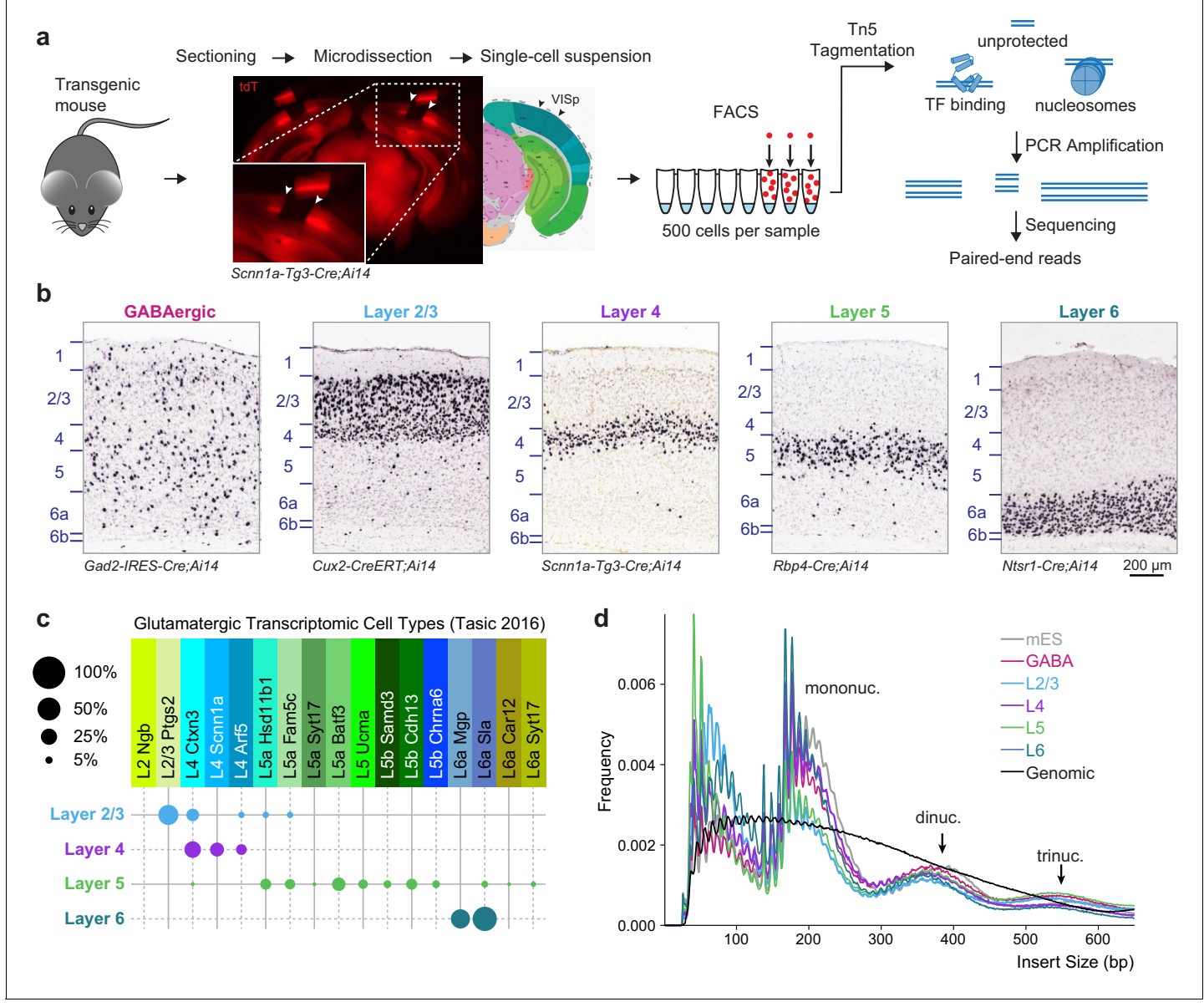

**Figure 1.** Overview of 500 cell ATAC-seq. (a) Mouse visual cortex was isolated from transgenic mice by brain sectioning and microdissection, and dissociated into single-cell suspension. 500 fluorescently labeled cells were isolated from the suspension by FACS, tagmented by Tn5, indexed and amplified by PCR, and sequenced on an Illumina platform. (b) Chromogenic RNA in situ hybridization (ISH) for *tdTomato* mRNA in Cre lines used for this study. Scale bar below Layer 6 applies to all panels. (c) Cell-type specificity of the glutamatergic Cre lines based on scRNA-seq profiling. Each Cre line labels at least two related transcriptomic types, with minimal overlap between Cre lines. Disc sizes are scaled by area to represent the percent of cells from each Cre line that were identified as each transcriptomic cell type. (d) Insert size frequency of ATAC-seq fragments from primary neurons reveals protection of DNA by individual nucleosomes and nucleosome multimers that is absent from purified genomic DNA sample (black line).

The following source data and figure supplement are available for figure 1:

**Source data 1.** Cre-line cell type composition table, as plotted in *Figure 1C*.

**Source data 2.** Fragment size frequencies for single replicates of each cell class.

**Figure supplement 1.** Quality control plots for ATAC-seq libraries.

max: 241 M, *Supplementary file 1A*), yielding >3 million unique, unambiguous fragments per replicate (min: 3.29 M, median: 6.9 M, max: 16.1 M, *Supplementary file 1A*). Each sample showed fragment size patterns characteristic of open chromatin and mono-, di-, and tri-nucleosomal fragments (*Figure 1D*, *Figure 1—figure supplement 1*), as well as a clear accessibility footprint around motifs for the ubiquitously-expressed transcription factor ATF2 throughout the genome (*Figure 1—figure supplement 1*). In comparison, data obtained from Tn5 tagmentation of purified genomic DNA from 500 mES cells do not display these characteristic signatures (*Figure 1D*, *Figure 1—figure supplement 1*).

We next identified accessible sites throughout the genome (*Supplementary file 1B*, Materials and methods), and then assigned each ATAC-seq peak to the nearest gene based on annotated transcription start sites (TSS). We call these peaks positionally associated with the nearby gene. This simple peak-gene association rule based on the gene nearest to the peak may not be accurate in all cases. Chromatin assumes cell-type specific conformation in the nucleus that may bring elements that are distant in linear DNA sequence or even on separate chromosomes together in space (*Dekker et al., 2013*). Therefore, our approach may misassign some peaks to genes that are closest to them based on the linear sequence of the genome. To assess if there are common patterns of peak positioning relative to TSS, we analyzed the distribution of ATAC-seq peak positions relative to genome-wide transcription start site (TSS) annotations (*Figure 2A*, *Figure 2—figure supplement 1*). The distribution of accessible sites revealed six peak populations: distal upstream (> 2 kb upstream of TSS; abbreviated −3), upstream (< 2 kb upstream, > 200 bp upstream; −2), proximal upstream (< 200 bp from TSS; −1), proximal downstream (< 200 bp; +1), downstream (200 bp to 2 kb; +2), and distal downstream (> 2 kb; +3). We found very similar distributions of peak locations in different cell classes, and found our distribution to be very similar to the one derived from previously published neuronal ATAC-seq data (*Camk2a-Cre*, [*Mo et al., 2015*]), *Figure 2A*, *Figure 2—figure supplement 1*). In addition, this comparison shows that our data, derived from 500 cells per sample, compare well with ATAC-seq data obtained from more than 1 million *Camk2a-Cre*-labeled nuclei per sample (*Mo et al., 2015*).

To assign putative function to ATAC-seq peaks, we compared our peak locations to the locations of histone modifications defined by ChIP-seq on *Camk2a-Cre*-labeled glutamatergic cells from the same study (*Mo et al., 2015*). In this study, four histone modifications were used to define promoters (H3K4me3), enhancers (H3K4me1 and H3K27ac), and polycomb-repressed chromatin (H3K27me3; *Figure 2B*). We found that most proximal peaks overlapped with promoter marks in all cell classes including mES cells, suggesting that these are indeed promoters, and that promoter accessibility is frequently not cell class-specific. In contrast, distal peaks had stronger class-specific biases, as distal glutamatergic peaks more frequently overlapped with enhancer marks than distal peaks from interneuron cell classes (*Mo et al., 2015*). This observation is in agreement with the specificity of *Camk2a-Cre*, which broadly labels pyramidal cells in the adult cortex (*Mo et al., 2015*; *Tsien et al., 1996*). ATAC-seq peaks from GABAergic and mES cells more frequently overlapped with polycomb-repressed regions from *Camk2a-Cre*-labeled glutamatergic cells. In summary, these results suggest that enhancer accessibility corresponds to specific cell classes, whereas many promoters may have similar accessibility across all classes.

## Chromatin accessibility is correlated with cell class-specific transcription

We performed hierarchical clustering of ATAC-seq peak data to define overall similarities in chromatin accesibility landscapes among our Cre-driver defined cell classes. ATAC-seq peak sets obtained from replicates of each cell class were most strongly correlated with each other (*Figure 3A*, Materials and methods). In addition, peak sets from these cell classes clustered according to previous transcriptomic findings (*Tasic et al., 2016*): GABAergic cells differed strongly from glutamatergic cell classes, and layer six was most distinct among glutamatergic cell classes (*Figure 3A*). When clustering was performed separately on TSS-proximal and TSS-distal peaks, we found that distal sites more cleanly delineated cell classes (*Figure 3—figure supplement 1*), as has been shown previously for hematopoieetic cell types (*Corces et al., 2016*). We identified differentially accessible peaks from each pairwise comparison between cell classes (*Supplementary file 1C*). Hierarchical clustering of these peaks showed diverse combinatorial accessibility patterns between different neuronal classes (*Figure 3B*, *Figure 3—figure supplement 2*), and examination of accessibility near known marker genes corresponded with expected class-specific chromatin state (*Figure 3C*).

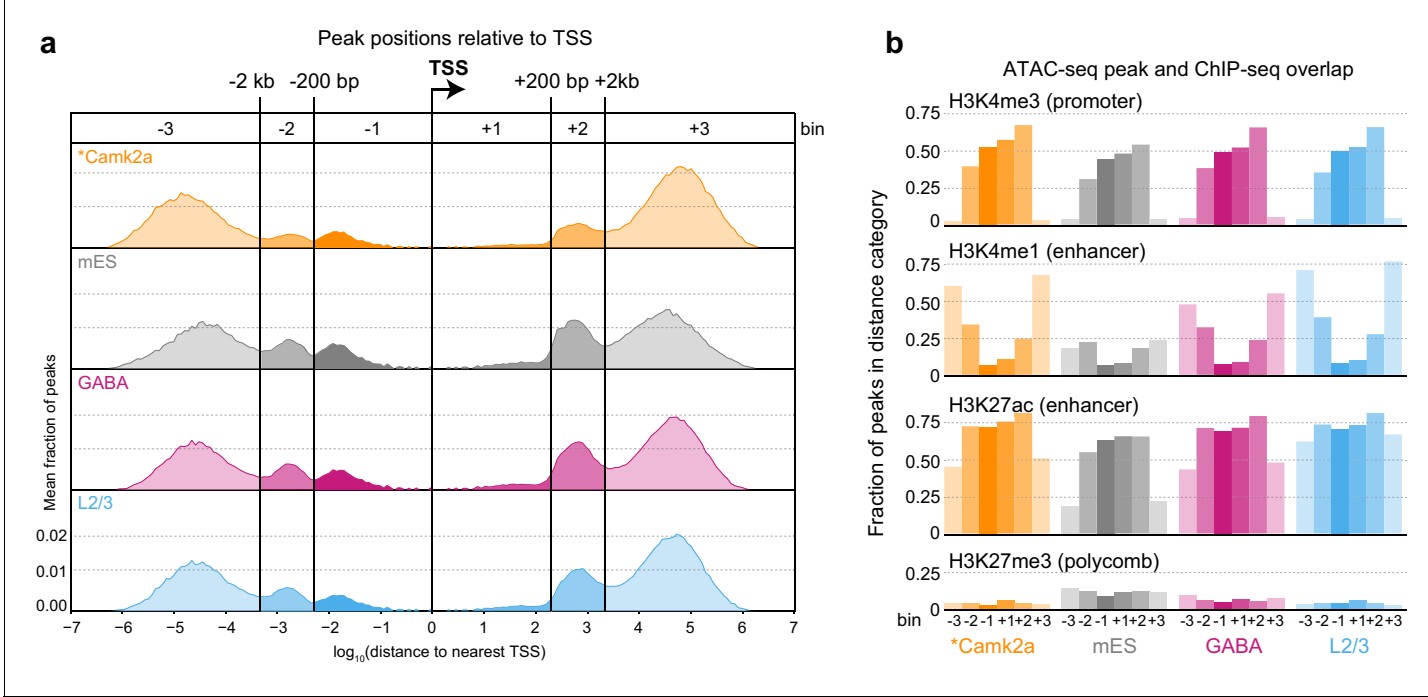

**Figure 2.** Peak locations relative to TSS and histone modifications. (a) Histogram of peak positions relative to the nearest TSS location. Distance to nearest TSS was used to group peaks into three upstream categories (−3, −2, and −1) and three downstream categories (+1, +2, and +3). (b) Fractions of ATAC-seq peaks in each distance category that overlap ChIP-seq peaks derived from *Camk2a-Cre* neurons show similar patterns of enrichment in excitatory types (Camk2a and L2/3), but reduced enhancer overlaps and increased polycomb-repressed region overlaps in mES and GABAergic cells. **\***, Camk2a data were from a previous study (***Mo et al., 2015***).

The following source data and figure supplement are available for figure 2:

**Source data 1.** Distributions of peak locations relative to TSS, used for ***Figure 2A***.
**Source data 2.** Histone modification frequencies for peaks by cell class and distance bin, used for ***Figure 2B***.
**Figure supplement 1.** Comparisons of ATAC-seq peaks to TSS locations and Camk2a-Cre-derived histone ChIP-seq data.

To place our data within the context of previously published datasets, we compared our peak sets to DNase I hypersensitivity (DHS) peaks from 14 tissues and ES cells in the Mouse ENCODE database (***Yue et al., 2014***). We found that our neuron-derived data clustered with 'Whole Brain' and 'Telencephalon' datasets, while our mES cell data clustered with 'ES-E14' datasets (***Yue et al., 2014***, ***Figure 3—figure supplement 3***). We also compared our data to ENCODE Whole Brain DHS data and to the cortical ATAC-seq data from glutamatergic (*Camk2a-Cre*) and GABAergic (*Pvalb-IRES-Cre* and *Vip-IRES-Cre*) cells (***Mo et al., 2015***, ***Figure 3—figure supplement 4***). Again, our data clustered as expected: our GABAergic datasets clustered with GABAergic whole cortex data, whereas our glutamatergic datasets clustered with the previously-published *Camk2a–Cre* datasets (***Figure 3—figure supplement 4***). We also find that a high fraction of our reads fall within HotSpot peaks (***Figure 3—figure supplements 5;*** 18–32%; similar to 24-37% for the previously published cortical ATAC-seq data).

Previous ATAC-seq and transcriptomic studies of neural tissue have shown that differential chromatin accessibility corresponds to differential gene expression (***Mo et al., 2016***, ***2015***). We have previously catalogued the transcriptomic types of cells in VISp by scRNA-seq, including cells from Cre lines used in this study (***Tasic et al., 2016***). To examine the correspondence between chromatin accessibility and gene expression, we assessed the correlation between differentially accessible peaks and differentially expressed genes for each pair of Cre lines (***Figure 4A***). For example, in the comparison between GABAergic and L2/3 cells, we selected all genes that were differentially

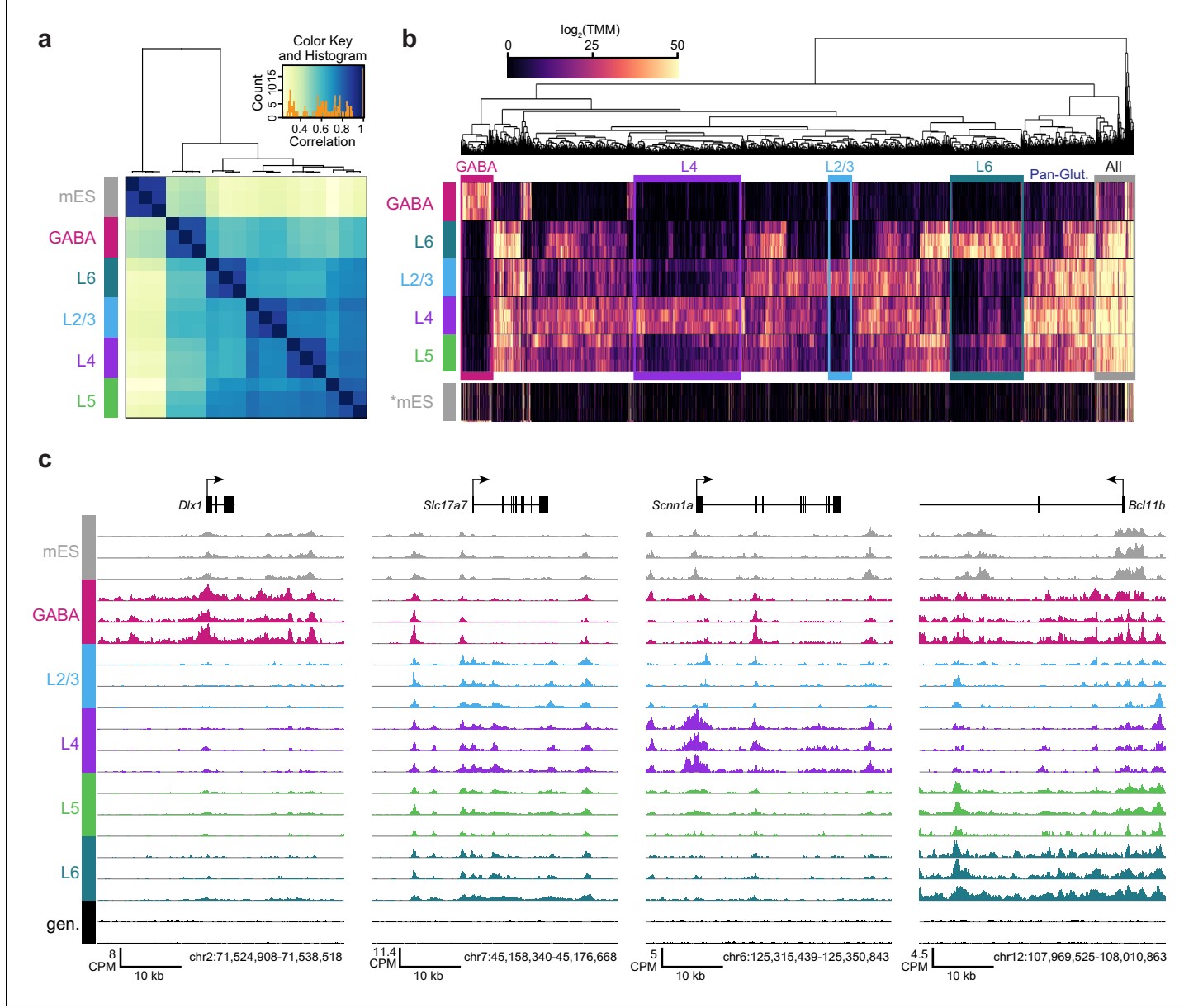

**Figure 3.** ATAC-seq samples cluster by cell class and reveal class-specific chromatin accessibility. (a) Correlation between each sample pair was based on the number of overlapping HotSpot regions weighted by normalized accessibility scores for each sample. The pairwise correlation scores were then used for hierarchical clustering (DiffBind). (b) Hierarchical clustering by complete linkage of the 7500 most statistically significant differentially accessible peaks as defined by DiffBind. Boxes highlight peak clusters with high cell-class specificity. *, Peak values from mES cells were arranged based on clustering of neural cell classes, but were not used for clustering. (c) Genomic regions near four marker genes show differences in accessibility across different cell classes: *Dlx1*, distal-less homeobox 1, is expressed in GABAergic cells; *Slc17a7*, solute carrier family 17 (sodium-dependent inorganic phosphate cotransporter), member 7, is expressed in glutamatergic cells; *Scnn1a*, sodium channel, nonvoltage-gated 1 alpha, is primarily expressed in L4 glutamatergic cells; *Bcl11b*, B-cell leukemia/lymphoma 11B, is strongly expressed in L5 and L6 cells, as well as in a subset of GABAergic cells. CPM, counts of overlapping fragments per million; gen, purified genomic DNA control.

The following figure supplements are available for figure 3:

**Figure supplement 1.** Hierarchical clustering of samples based on TSS-proximal or TSS-distal HotSpot results.

**Figure supplement 2.** Intersections among peak sets derived from different cell classes.

**Figure supplement 3.** Correlation of HotSpot peak sets from this study and ENCODE tissue DNase-seq.

*Figure 3 continued on next page*

*Figure 3 continued*

**Figure supplement 4.** Correlation of brain-derived DNase-seq and ATAC-seq datasets.

**Figure supplement 5.** Fraction of Reads in Peaks.

expressed between this pair of cell classes, then separated this gene set into those with higher expression in GABAergic cells, and those with higher expression in L2/3. We then surveyed the accessibility scores of all peaks that were positionally associated with these two gene sets in both GABAergic and L2/3 cells (box plots, *Figure 4A*). We found that overall peak accessibility corresponded to differential gene expression (*Figure 4A*): 83–94% of differentially accessible peaks, which were associated with differentially expressed genes, were positively correlated with gene expression (*Figure 4—figure supplement 1*).

The general pattern of higher chromatin accessibility corresponding to higher gene expression holds true for individual marker genes that distinguished transcriptomic cell classes. For example, genes that are more highly expressed in GABAergic cells than in L4 cells, including *Gad1*, *Slc6a1*, and *Dlx5*, display prominent peaks of chromatin accessibility in GABAergic cells and not in L4 cells. Likewise, *Slc17a7*, *Bdnf*, and *Nrn1*, which are more highly expressed in L4 than in GABAergic cells, are associated with more accessible chromatin peaks specifically in L4 cells (*Figure 4B*). Similarly, we find regions near differentially expressed genes that are associated with layer-specific gene expression, such as *Calb1*, *Pou3f2*, and *Rorb*, which are expressed in L4, and *Bcl11b*, *Nfia*, and *Nos1ap*, which are expressed in L6 (*Figure 4C*). In a minority of the cases, we also detect strongly differentially-accessible sites that are anti-correlated with gene expression. Such is the case for *Hkdc1*, which is more highly expressed in L4, but the highlighted peak in *Figure 4C* is significantly more accessible in L6 cells. These sites are potentially associated with binding of transcriptional repressors, as has been shown for NRSF/REST (*Thurman et al., 2012*). Alternatively, these peaks may be misassigned due to the nearest-TSS peak-gene association rule used in our analysis. We find that negatively correlated peaks make up 6–13% of the differentially accessible peaks that are associated with differentially accessible genes (*Figure 4—figure supplement 1*).

## Layer-specific transcription factor motifs are enriched in ATAC-seq peak clusters

We identified modules of peaks with shared patterns of accessibility across the four glutamatergic cell classes using K-means clustering of differentially accessible peaks and differentially expressed genes (*Figure 5A*). Briefly, we first clustered peaks and genes separately, then selected common patterns to generate a merged set of binary cluster centers that were used to build modules of common accessibility and gene expression (Materials and methods, *Figure 5—figure supplement 1*, *Supplementary file 1C*). We found modules that are specific to each layer, but also sets of layers: Upper+, L2/3 and L4; Lower+, L5 and L6; L4-absent; and L6−absent.

After building peak and gene modules, we tested if the peak locations in each module were statistically significantly associated with genes in gene expression modules. We found that each peak module contained a significantly higher number of peaks positionally associated with gene modules with similar patterns of layer specificity (along the diagonal of *Figure 5B*, *Figure 5—figure supplement 2*). We also found associations between peak modules and gene modules that had similar but non-identical patterns. For example, the Lower+ peak module was significantly associated with genes that are highly expressed only in L6 (*Figure 5B*).

We next looked for TFs that may be responsible for layer-specific transcription by searching for differentially-enriched motifs between our peak modules using Analysis of Motif Enrichment (AME, *Figure 6A*). For this analysis, we relied on a JASPAR TF motif database (Materials and methods), and thus our motif searches carry the biases of database-driven TF motif search analyses – databases may be incomplete, and depend on previous studies. Despite this caveat, we were able to distinguish many differentially enriched motifs by comparing each of the peak accessibility modules to peak modules with dissimilar patterns of accessibility. For example, peaks in the L2/3+ module were

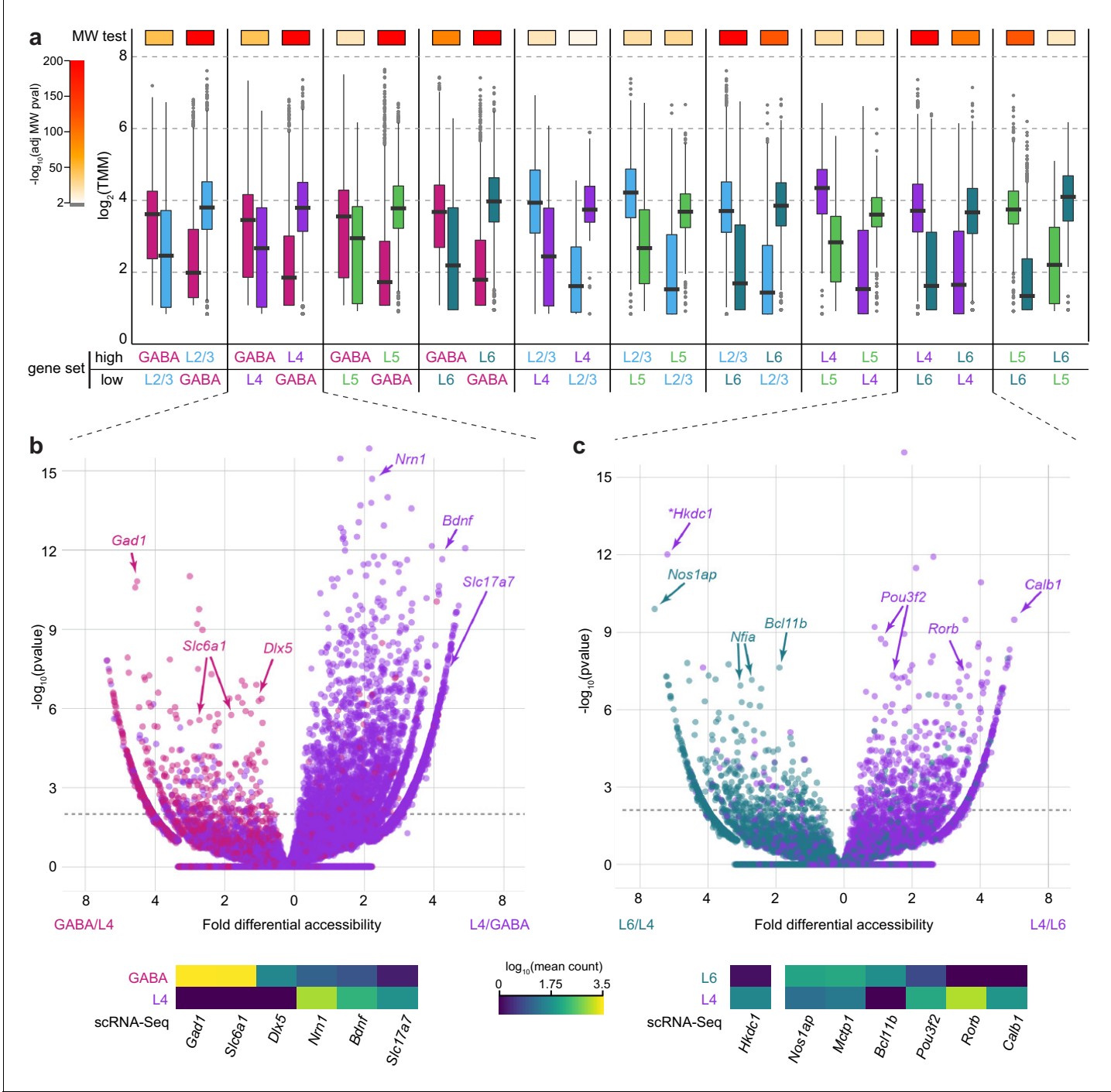

**Figure 4.** Chromatin accessibility corresponds to cell class-specific transcription. (a) For each pair of cell classes, we identified differentially expressed genes (adjusted p-value < 0.05 and fold change > 2), then separated genes into two groups based on the class with higher expression. We then asked if the peaks that are positionally associated with each group of genes had higher accessibility in the class with higher gene expression. For each peak set, box plots show the median accessibility (black bar), quartiles (boxes), 1.5 × interquartile range (whiskers), and outliers (points). The two distributions of peak accessibility scores for each gene set were compared using a Mann-Whitney U test (MW test). Adjusted p-values are displayed using heatmap boxes above each gene set. (b, c) Two example volcano plots for pairwise comparisons between GABAergic and L4 cells (b) and L4 and L6 cells (c) showing all peaks associated with differentially-expressed genes (adjusted p-value < 1×10$^{-6}$) . Peaks associated with select marker genes are labeled, and the corresponding average gene expression for these genes from single-cell RNA-seq data for each cell class is shown below. *, *Hkdc1*-associated peak is more accessible in L6 cell class, although expression of *Hkdc1* is greater in L4 class.

*Figure 4 continued on next page*

*Figure 4 continued*

The following source data and figure supplements are available for figure 4:

**Source data 1.** Peak accessibility scores (TMM) for peaks associated with gene sets in *Figure 4A*.

**Source data 2.** Mann-Whitney test results for each comparison in *Figure 4A*.

**Source data 3.** Gene expression data for the heatmap at the bottom of *Figure 4B*.

**Source data 4.** Differential accessibility and $-\log_{10}$(pvalue) scores used to generate the volcano plot in *Figure 4B*.

**Source data 5.** Gene expression data for the heatmap at the bottom of *Figure 4C*.

**Source data 6.** Differential accessibility and $-\log_{10}$(pvalue) scores used to generate the volcano plot in *Figure 4C*.

**Figure supplement 1.** Pairwise comparisons of peak accessibility for peaks associated with differentially-expressed genes.

**Figure supplement 2.** Permutation of peak-gene associations.

contrasted with peaks from L4+, L5+, L6+ and Lower+ modules, but not Upper+, L4−, or L6−, which include many peaks that show high accessibility in L2/3 (Materials and methods, *Figure 6—figure supplement 1*). This analysis yielded a set of enriched TF binding site motif families that we used for downstream analysis: CUX, DLX, EGR1, FOS, FOXP, MEF2, MEIS, NEUROD, NFIA, POU3F, RFX3, RORB, and TBR1. For each module, we were able to identify at least one significantly enriched motif family. In layer 2/3 cells, we detect high enrichment of CUX, EGR1, FOXP, MEF2, POU3F, and RFX. In L4, we found enrichment of RORB and NEUROD motifs. Layers 4 and 5 both show enrichment of FOS motifs. L6 cells have a very different profile of enriched TF motifs, with depletion of many of the factors listed above, but enrichment of CUX, MEIS, NFIA and TBR1 motifs.

To identify specific transcription factors that may be driving accessibility at these sites, we examined the average gene expression of candidate TFs in each glutamatergic cell class. We used the TreeFam database (*Ruan et al., 2008*) to identify closely related TFs which may bind similar motifs. We then removed genes with low expression in the single cell RNA-seq dataset, and those previously found not to bind the motifs in our motif sets (*Figure 6B*, *Figure 6—figure supplement 2*, Materials and methods). For most of the motifs differentially enriched in the peak modules, we find at least one corresponding TF family member with correlated or anticorrelated differential gene expression. For example, enrichment of accessible ROR motifs is positively correlated with *Rorb* expression in L4. In L2/3, we see strong enrichment of RFX3 motifs, which corresponds to high *Rfx3* and *Rfx7* expression in L2/3. Conversely, open FOXP motifs are enriched in upper layers (enriched in L2/3+, Upper+ and L6− modules; depleted in L6+ and Lower+), while *Foxp2* is most highly expressed in L6. This is in agreement with the previously reported repressive function of FOXP2, which has been shown to recruit histone deacetylase complexes (*Chokas et al., 2010*). Similarly, CUX motif enrichment is inversely correlated with *Cux1* and *Cux2* expression, consistent with the role of these TFs as repressors (*Li et al., 2010*).

We also examined the accessibility of the motifs with a single base-pair resolution (TF footprinting) using Tn5 transposase insertion sites, as has been done previously with ATAC-seq (*Mo et al., 2015*) and similarly with DNase-seq (*Vierstra and Stamatoyannopoulos, 2016*). The FOXP, NEUROD, and RFX motifs found in all peaks in our dataset displayed expected patterns of accessibility that corresponded to the enrichment of motifs within layer-specific modules (*Figure 6C*): FOXP motifs are most accessible on L2/3, L4, and L5, with the least insertions in L6, where *Foxp2* may repress chromatin accessibility at FOXP motifs. In L4, NEUROD motifs were significantly more accessible than in L2/3, L5, or L6, which may be driven by coexpression of *Neurod1* and *Neurod6* (*Figure 6B*). For RFX motifs, we see the most Tn5 insertion sites in L2/3 cells than in other layers, possibly due to expression of *Rfx1* and *Rfx3* in L2/3 (*Figure 6B*).

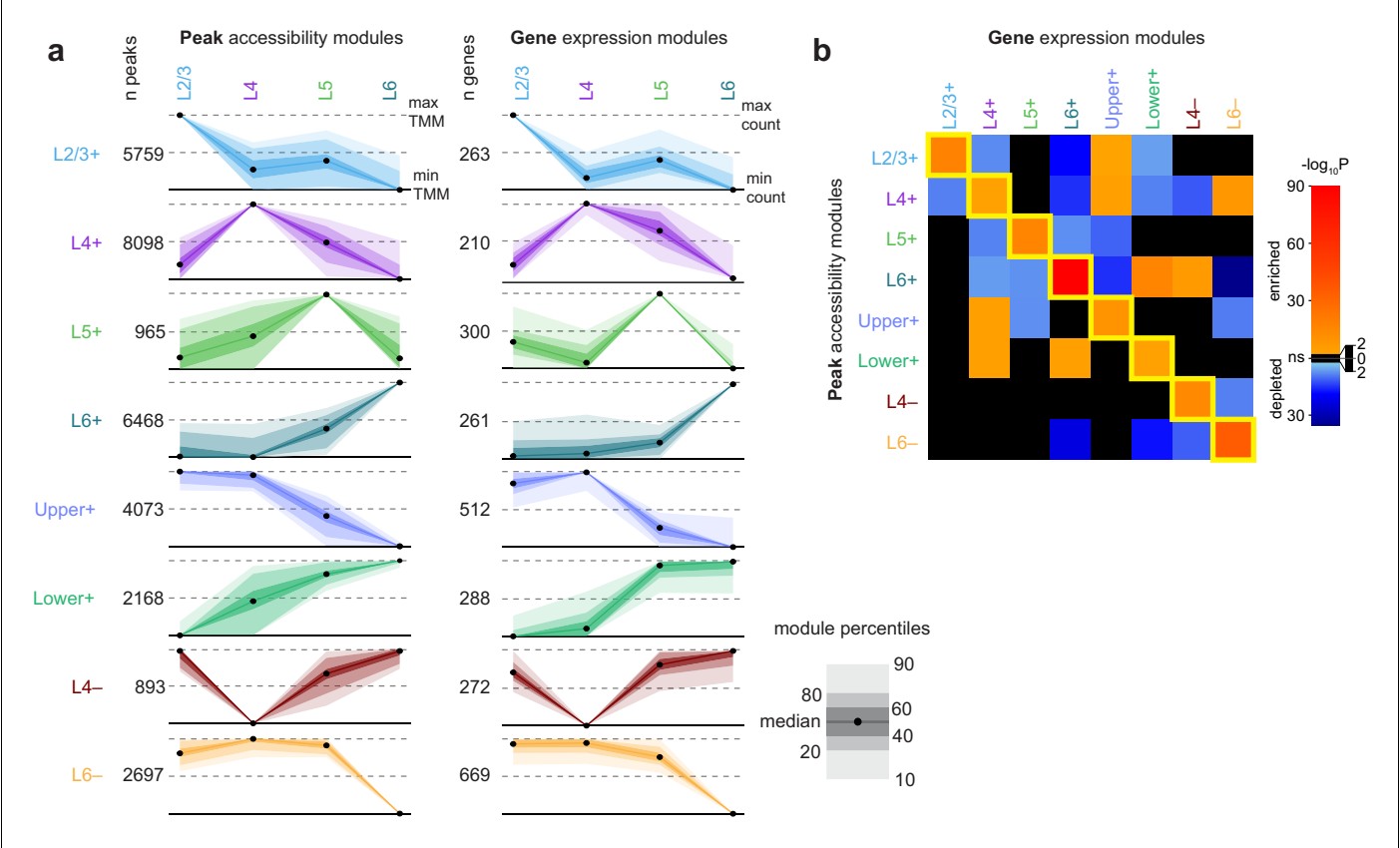

**Figure 5.** Clustering of peaks and genes reveals common patterns of chromatin accessibility and gene expression. (**a**) Scaled module profiles derived from *k*-means clustering of peaks and genes (Materials and methods). Points represent median values, and shaded areas represent percentiles as shown in the legend. (**b**) We calculated how frequently peaks in each peak module were positionally-associated with genes in each gene module, then computed enrichment or depletion using Fisher's exact tests for enrichment. The heatmap represents the log-transformed, adjusted p-values from Fisher's exact tests, with enrichment (odds ratio > 1) in red and depletion (odds ratio < 1) in blue. Black indicates non-significant enrichment or depletion (adjusted p-value > 0.01).

The following source data and figure supplements are available for figure 5:

**Source data 1.** Fisher's exact test result values presented in *Figure 5B*.

**Source data 2.** Quantile values for gene clusters presented in *Figure 5A*.

**Source data 3.** Quantile values for peak clusters presented in *Figure 5A*.

**Figure supplement 1.** Example instance of initial *k*-means clustering of differentially accessible peaks and differentially expressed genes.

**Figure supplement 2.** Log-odds ratios for tests of Peak-Gene module association.

## Transcription factor networks underlying layer-specific transcriptomes

We next sought to identify interactions between key TFs that may underlie layer-specific transcrip-tomes (*Figure 7*). For this analysis, we focused on the targets of 12 highly differentially expressed TFs: *Cux1, Egr1, Fos, Mef2c, Neurod6, Pou3f2, Rfx3,* and *Rorb*, which are highly expressed in upper layers; and *Foxp2, Meis2, Nfia,* and *Tbr1*, which are highly expressed in lower layers (*Figure 6B*). We then designated each TF as a likely activator (*Egr1, Fos, Mef2c, Nfia, Neurod6, Pou3f2, Rfx3, Rorb, Meis2,* and *Tbr1*) or repressor (*Foxp2* and *Cux1*). A full schematic for the selection of these putative key regulators is provided in *Figure 7—figure supplement 1*. Next, we searched our peaks for dif-ferentially enriched motifs that are putative targets of these TFs. For each motif, we applied filtering

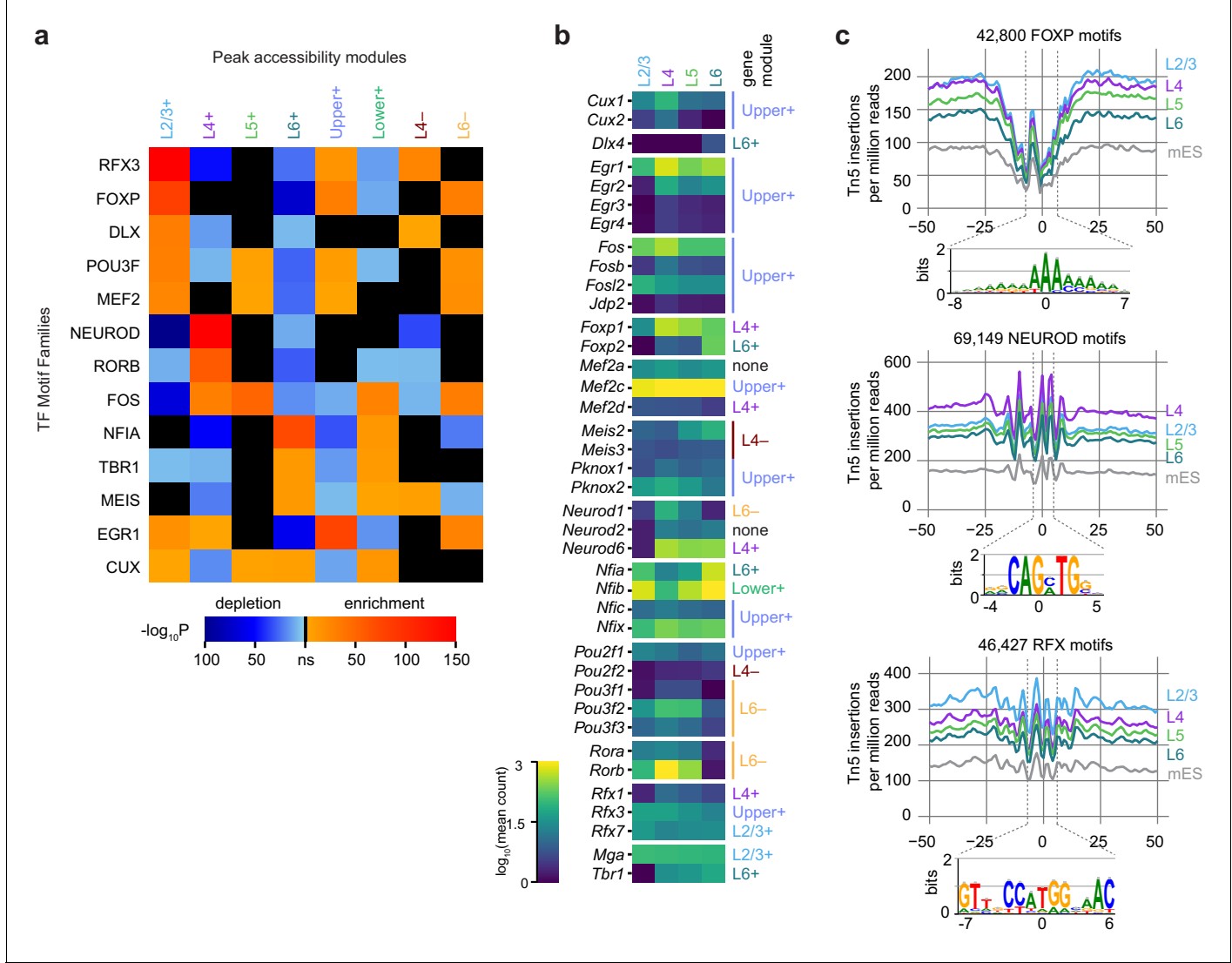

**Figure 6.** Peak module analysis reveals layer-specific enrichment of transcription factor motifs. (**a**) Select TF motif families are significantly enriched or depleted in specific peak modules. Enrichment and depletion were calculated relative to unrelated modules (*Figure 6—figure supplement 1*) using AME. (**b**) Expression of genes belonging to TF families identified by Treefam and other criteria (Materials and methods). Heatmap shows mean gene expression counts within each cell class based on single-cell RNA-seq data (Materials and methods). For each gene, the most strongly correlated gene module is listed to the right of the heatmap. (**c**) Tn5 footprinting for select motif families in each glutamatergic cell class and mES cells. The cut site frequency was calculated at each base position relative to the center of each corresponding motif . Frequencies are shown as the number of Tn5 insertions (locations of 5′ ends of fragments) per million reads.

The following source data and figure supplements are available for figure 6:

**Source data 1.** AME result p-values, as plotted in *Figure 6A*.

**Source data 2.** Gene expression values used for *Figure 6B*.

**Source data 3.** FOXP motif Tn5 insertion frequency data.

**Source data 4.** NEUROD motif Tn5 insertion frequency data.

**Source data 5.** RFX motif Tn5 insertion frequency data.

**Figure supplement 1.** Background set selection for AME and top significant AME results for each peak module.

*Figure 6 continued on next page*

*Figure 6 continued*

**Figure supplement 2.** Transcription factor gene expression in individual cells arranged by cell class.

criteria to find putative regulatory targets based on motif enrichment in peak modules, association of peak module with a gene module, differential accessibility of the peak, differential expression of the nearest gene, and correlation with expression of the associated TF (Materials and methods and *Figure 7—figure supplement 1*). We then built a network using our 12 key regulators as nodes, and linked the nodes using their putative target motifs as directional edges (from the regulating factor

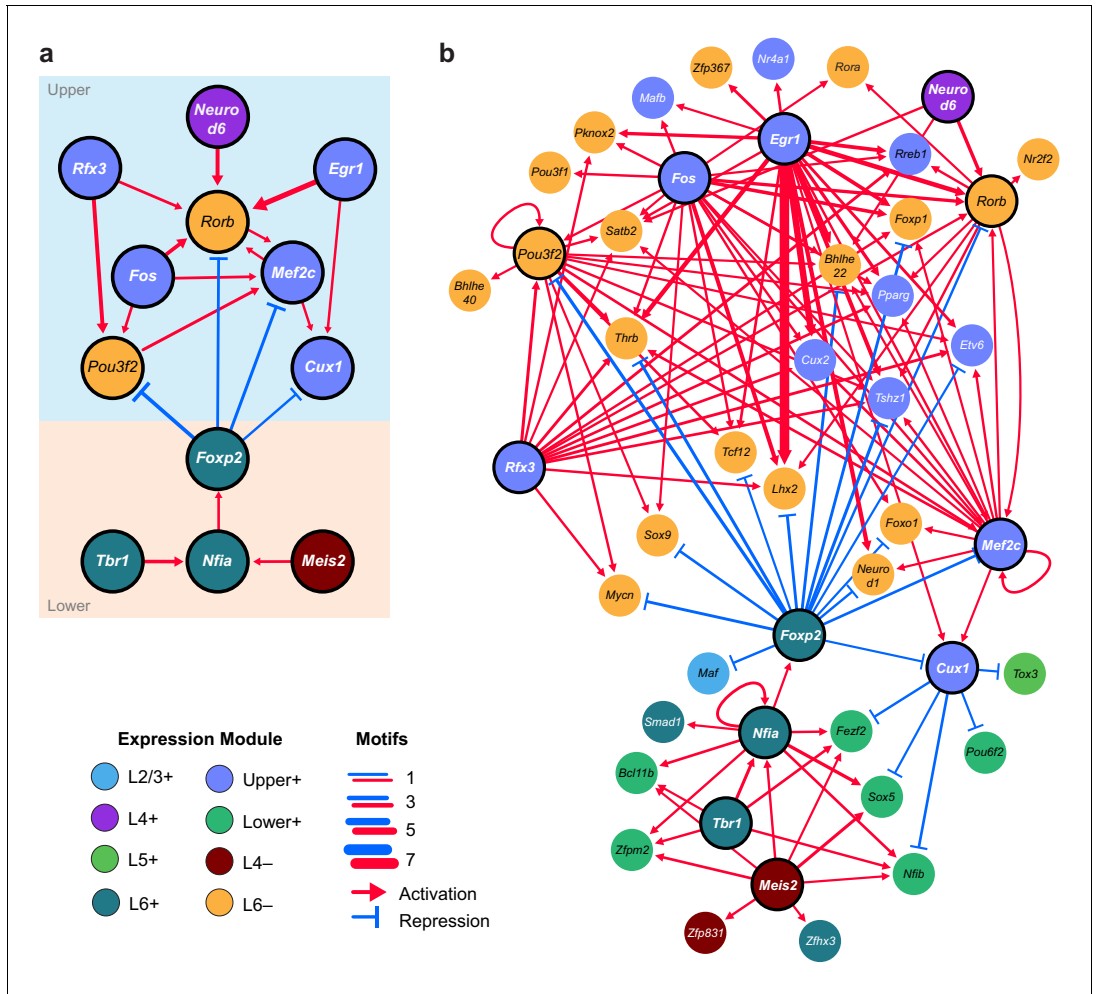

**Figure 7.** Putative regulatory interactions that govern layer-specific chromatin and transcriptomic state in glutamatergic cell classes. (**a**) Putative regulatory interactions between key TFs. (**b**) Putative regulatory interactions between key TFs and other differentially expressed TFs in glutamatergic cell classes. Key TFs have bold outlines, whereas targets have no outline. Each TF node is colored according to the most strongly correlated gene module. Edges of the network represent differentially-accessible motifs, and are weighted based on the number of motifs observed near the target gene.

The following source data and figure supplement are available for figure 7:

**Source data 1.** Data used to build the network presented in *Figure 7B* and *Figure 8*.

**Figure supplement 1.** Flowcharts for selecting TFs and TF targets to construct a network of regulatory interactions.

to its target motif, *Figure 7A*). Expanding the network to include additional targets that are TF genes outside of the set of 12 key regulators reveals additional regulatory interactions that may be important for maintenance of layer-specific transcriptomes (*Figure 7B*). These networks show only interactions that were strongly coherent with our peak and gene module analysis (Materials and methods). The relative transcription levels of each of the genes in the resulting network show a sharp division between upper (L2-3 and L4) and lower (L5 and L6) layers (*Figure 8*).

Central to the upper/lower network division is the Forkhead-box TF *Foxp2*, which targets many of the upper-layer transcription factor genes, including *Rorb* and *Pou3f2*. *Foxp2* is an important gene for development of speech in humans (*Becker et al., 2015*), and vocalization in mouse (*Castellucci et al., 2016*) and songbirds (*Chen et al., 2013*; *Murugan et al., 2013*). *Foxp2* also plays key roles in neural differentiation (*Chiu et al., 2014*; *Tsui et al., 2013*) and neurite outgrowth during development (*Vernes et al., 2011*). Both FOXP1 and FOXP2 have been shown to interact with the NuRD chromatin remodeling complex, which includes the histone deacetylases HDAC1 and HDAC2 (*Chokas et al., 2010*). Recruitment of NuRD to FOXP target motifs by FOXP1 or FOXP2 is thought to create a repressed chromatin state through deacetylation of histones. Although we observe expression of *Foxp1* across upper layers (L2/3-L5, *Figure 6B*), we see a significant decrease of FOXP motif accessibility in L6, which specifically expresses *Foxp2* (*Figure 6A,B*). Thus, we have assigned the FOXP motifs that are less accessible in L6 as putative FOXP2 targets in our network.

The repressor CUX1 has several targets among lower-layer transcription factors, and may play a similar repressive role in upper-layer cell classes as FOXP2 in lower-layer cell classes. *Cux1* encodes a member of the Cut-like homeobox family of transcriptional repressors, which also includes *Cux2* (*Quaggin et al., 1996*). In the network diagram, we attributed CUX motif targets to CUX1 due to higher average expression of *Cux1* mRNA compared to *Cux2* mRNA in upper layers (*Figure 6B*) although CUX2 may also regulate some of these targets, especially in L2/3, where *Cux2* expression is highest. *Cux1* and *Cux2* expression is most strongly correlated with the Upper+ gene expression module in this study, and both of these TFs have been shown to regulate dendritic branching and morphology in upper cortical layers (*Cubelos et al., 2010*; *Li et al., 2010*). We see a depletion of CUX motifs where *Cux1* is most highly expressed (L4+ and Upper+ peak clusters, *Figure 6A*), so have assigned inversely-correlated CUX-containing peaks to *Cux1*. *Cux1* targets include *Nfib*, *Fezf2*, *Pou6f2*, and *Sox5*, all of which are transcriptional regulators that are highly expressed in lower layers (*Figure 6—figure supplement 2*). Intriguingly, we find a FOXP binding site in a peak associated

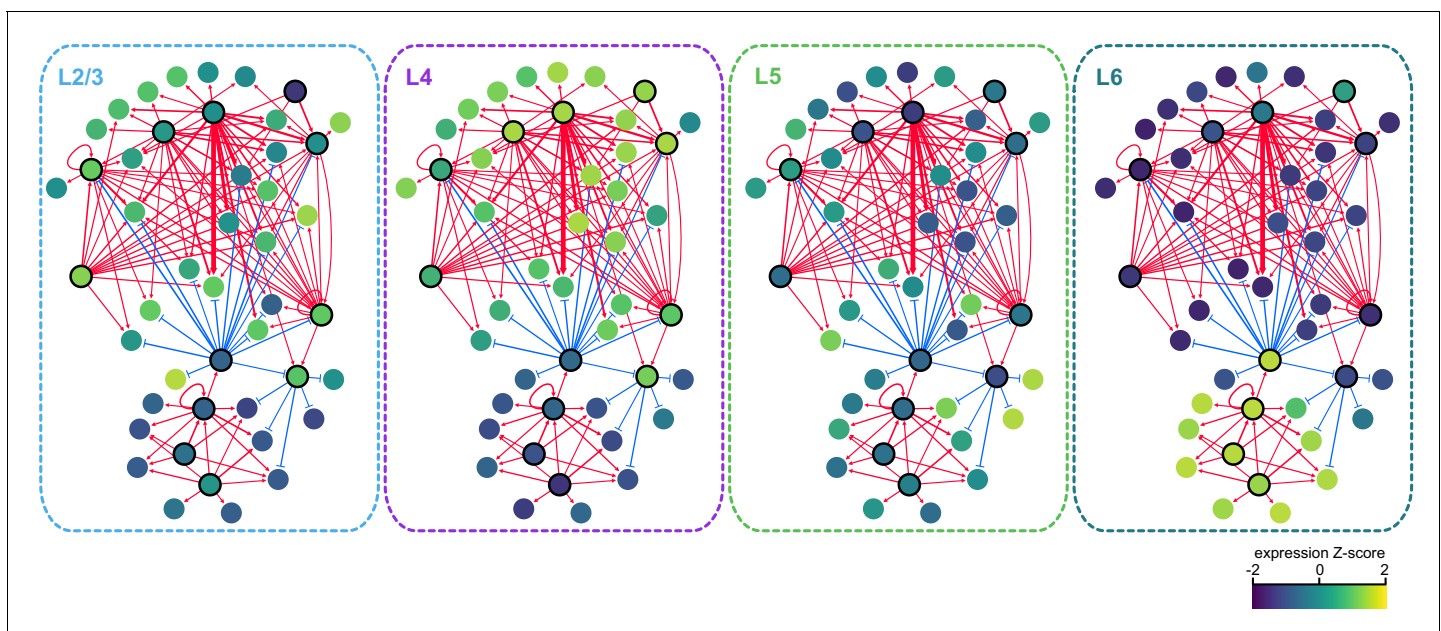

**Figure 8.** Gene expression patterns of layer-specific transcription factors. The location and identity of each node is the same as in the regulatory network presented in *Figure 7*. The color of each node represents the normalized, average gene expression across all cells in each cell class.

with the *Cux1* gene. FOXP2 binding to this site may repress *Cux1*, and relieve CUX1-mediated repression of its targets in lower cortical layers (*Figure 7B*).

We also find several key transcriptional activators that may be responsible for differential expression of layer-specific genes. T-box brain gene 1 (*Tbr1*) and nuclear factor I/A (*Nfia*) have many targets among other transcription factors that are expressed in lower layers, including *Foxp2*, *Fezf2*, *Zfpm2*, *Bcl11b*, and *Pdlim1* (*Figure 7*). Loss of *Nfia* in humans and mice has been shown to disrupt corpus callosum development (*das Neves et al., 1999*; *Lu et al., 2007*). *Nfia* is important for balancing gliogenesis of oligodendrocytes and astrocytes (*Glasgow et al., 2014*), and is a key factor for the onset of spinal cord gliogenesis (*Deneen et al., 2006*). Other NFI transcription factors are also expressed in visual cortex. However, *Nfic* and *Nfix* expression is not correlated with NFIA motif enrichment, and *Nfib* is more broadly expressed in L2/3 and L5 than *Nfia* (*Figure 6B*, *Figure 6—figure supplement 2*). Thus, we have assigned NFIA motifs to *Nfia*. TBR1 binding sites are found near *Nfia*, suggesting that TBR1 may regulate *Nfia* in addition to other target TFs. *Tbr1* has been associated with autism spectrum disorders (*Huang and Hsueh, 2015*), regulates corticofugal cell identities during development (*McKenna et al., 2011*), and represses target genes including *Fezf2* during development of the corticospinal tract (*Han et al., 2011*). However, previously published scRNA-seq data show that *Tbr1* and *Fezf2* are co-expressed in some L5 and L6a transcriptomic cell types (*Tasic et al., 2016*), and our network analysis suggests that TBR1 may activate *Fezf2* in adult neurons in lower cortical layers (*Figure 7B*, *Figure 6—figure supplement 1*).

In the upper layers, POU3F2, RORB, and RFX3 appear to play central roles in regulation of many other TFs that are expressed in L2/3 and L4 classes. Regulatory factor X3 (*Rfx3*) is a key gene for guidance of thalamocortical axons to L2/3 and L4 (*Magnani et al., 2015*), as well as for development of the corpus callosum (*Benadiba et al., 2012*). POU domain, class 3, transcription factor 2 (*Pou3f2*), also known as *Brn2*, is essential for correct migration of upper-layer neurons in the developing cortex (*McEvilly et al., 2002*; *Oishi et al., 2016*). Retinoic acid-related orphan receptor B (*Rorb*), which is highly expressed in L4 of the visual cortex is important for barrel cluster development (*Jabaudon et al., 2012*) and L2/3 and L4 specification during cortical development (*Oishi et al., 2016*). Heterozygous deletion or loss of function of RORB in humans has been associated with epilepsy (*Baglietto et al., 2014*; *Rudolf et al., 2016*) and genetic variants associated with Rorb are associated with bipolar disorder (*Geoffroy et al., 2015*; *Lai et al., 2015*).

In our network, we also observe several NEUROD motifs in peaks that are positionally associated with *Rorb* (*Figure 7A*). Neurogenic differentiation 1 and 6 (*Neurod1* and *Neurod6*) are basic helix-loop-helix (bHLH) TFs, which form homo- and heterodimers. Thus, co-expression of *Neurod1* and *Neurod6* may be responsible for enrichment of NEUROD motifs in the L4+ peak module (*Figure 6A*).

## Putative class-specific regulatory sites near *Nfia* and *Cux1* may establish differential expression of L6 genes

To understand key interactions that control the differential transcription state of Layer 6 cells compared to upper layers, we examined the differential accessibility landscapes of *Nfia* and *Cux1*. *Nfia* is expressed almost exclusively in L6 cells (*Figure 9A*). We found 55 peaks that were positionally associated with *Nfia*, 14 of which were both significantly more accessible (DiffBind adjusted p-value < 0.01) and displayed more than 2-fold higher accessibility in L6 cells compared to L4 (*Figure 9B*). These peaks were present near a TSS specific to two *Nfia* isoforms (red arrow in *Figure 9C*), and in many downstream sites (*Figure 9C*). Examination of individual peaks shows accessible motifs in L6 that are not accessible in L4, (*Figure 9D*) including TBR motifs (numbered peaks 1, 2 and 4), NFIA motifs (peaks 3 and 4), and an RFX motif (peak 3), which overlaps the NFIA motif.

*Cux1* encodes a cut-like homeobox-family repressor that is strongly expressed in glutamatergic cells in upper, but not lower, cortical layers (*Figure 10A*). To define regulatory elements of *Cux1*, we examined differential accessibility of 18 *Cux1*-associated ATAC-seq peaks in the comparison between L4 and L6 classes (*Figure 10B*). We found five peaks that were differentially accessible between the two cell classes (p-value < 0.01), all of which were more accessible in L4 cells. These five sites are distal to the *Cux1* TSS, located in a region surrounding the second *Cux1* exon. In agreement with our general observations (*Figure 3—figure supplement 1*), the *Cux1* TSS appears accessible across all five Cre lines (*Figure 10C*). This suggests that the *Cux1* TSS is poised for transcription across all neural cell types, but *Cux1* expression is only achieved through the function

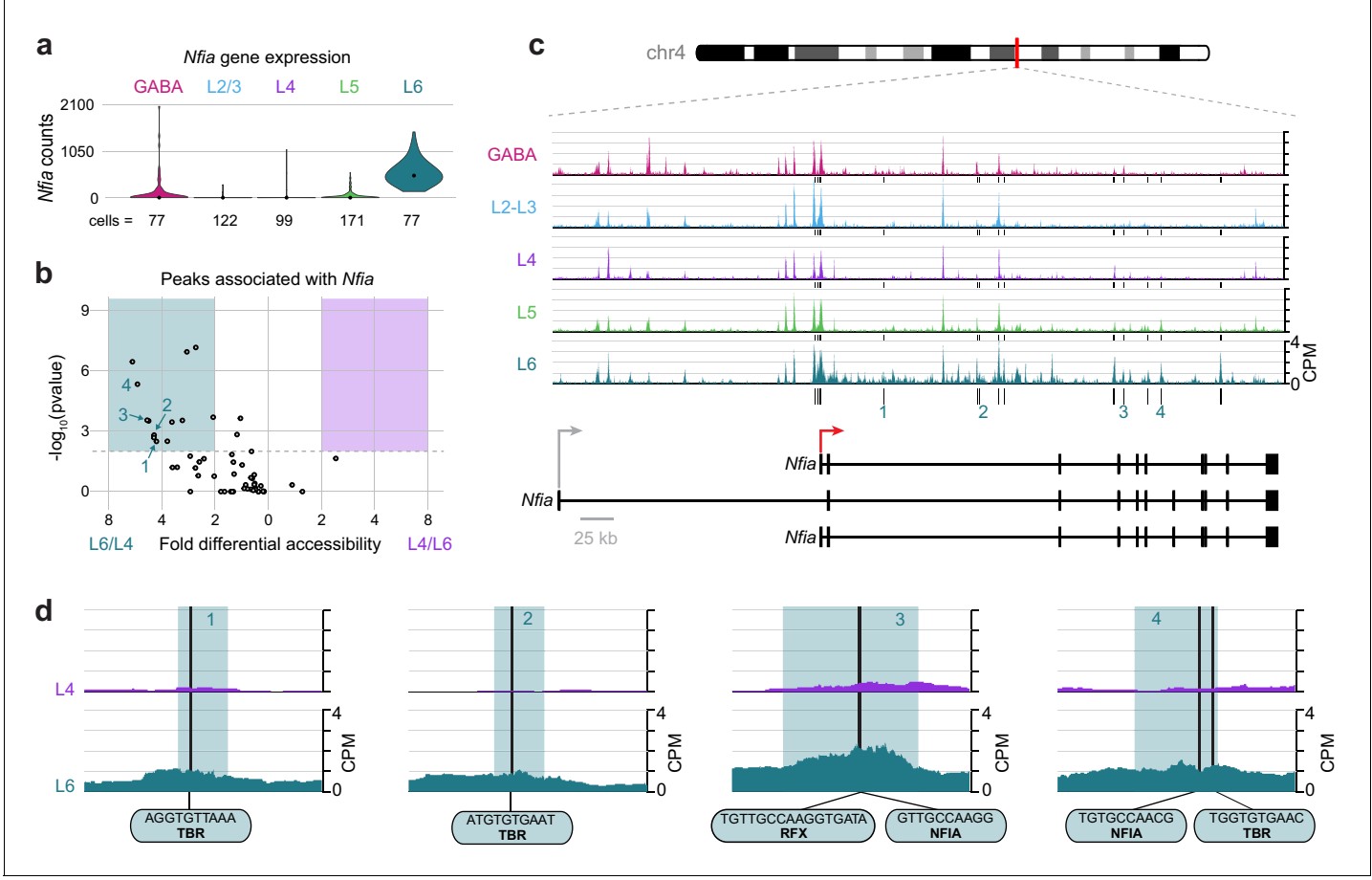

**Figure 9.** The Layer-specific regulatory landscape of *Nfia*. (**a**) *Nfia* gene expression distributions based on scRNA-seq in each neuronal cell class. (**b**) Volcano plot showing all peaks that are positionally associated with *Nfia* in L6 and L4. Significantly differentially accessible peaks (adjusted p-values < 0.01 and > 2-fold change in accessibility score) are highlighted as L6-specific (green box) or L4-specific (purple box). (**c**) Chromatin accessibility near the *Nfia* gene in each cell class (547 kb window; mm10 chr4:97,576,942–98,123,876). Vertical lines below the tracks represent the locations of peaks that are significantly more accessible in L6 compared to L4. (**d**) 1 kb windows centered on four numbered peaks that contain putative TF binding sites. TF motif locations within each peak are marked by black bars. CPM, counts of overlapping sequenced fragments at each position per million mapped reads for each cell class. All fragment overlap panels are plotted on the same scale (0 to 4 CPM).

The following source data is available for figure 9:

**Source data 1.** *Nfia* expression values used to generate the plot in *Figure 9A*.

**Source data 2.** Peak statistics for peaks positionally associated with *Nfia*, used to generate *Figure 9B*.

of these downstream, distal enhancers. In agreement with *Cux1* expression, L2/3 and L4 show the same accessibility pattern near *Cux1*. Individual peaks that are highly accessible in L4 but not L6 reveal interactions that may drive class-specific expression of *Cux1* in upper layers. Close examination of 4 of the 5 Cux1 peaks reveal NEUROD motifs are found in all 4 of the highlighted peaks; several FOS motifs (highlighted peaks 2 and 4); and single EGR (peak 2), FOXP (Peak 3), MEF2 (peak 4), and MEIS (peak 3) motifs near *Cux1* are also accessible in L4 neurons. The FOXP motif may recruit FOXP2 and NuRD complexes to repress *Cux1* transcription in lower layers, while EGR1/EGR3, MEF2C, and NEUROD may regulate expression in upper layers.

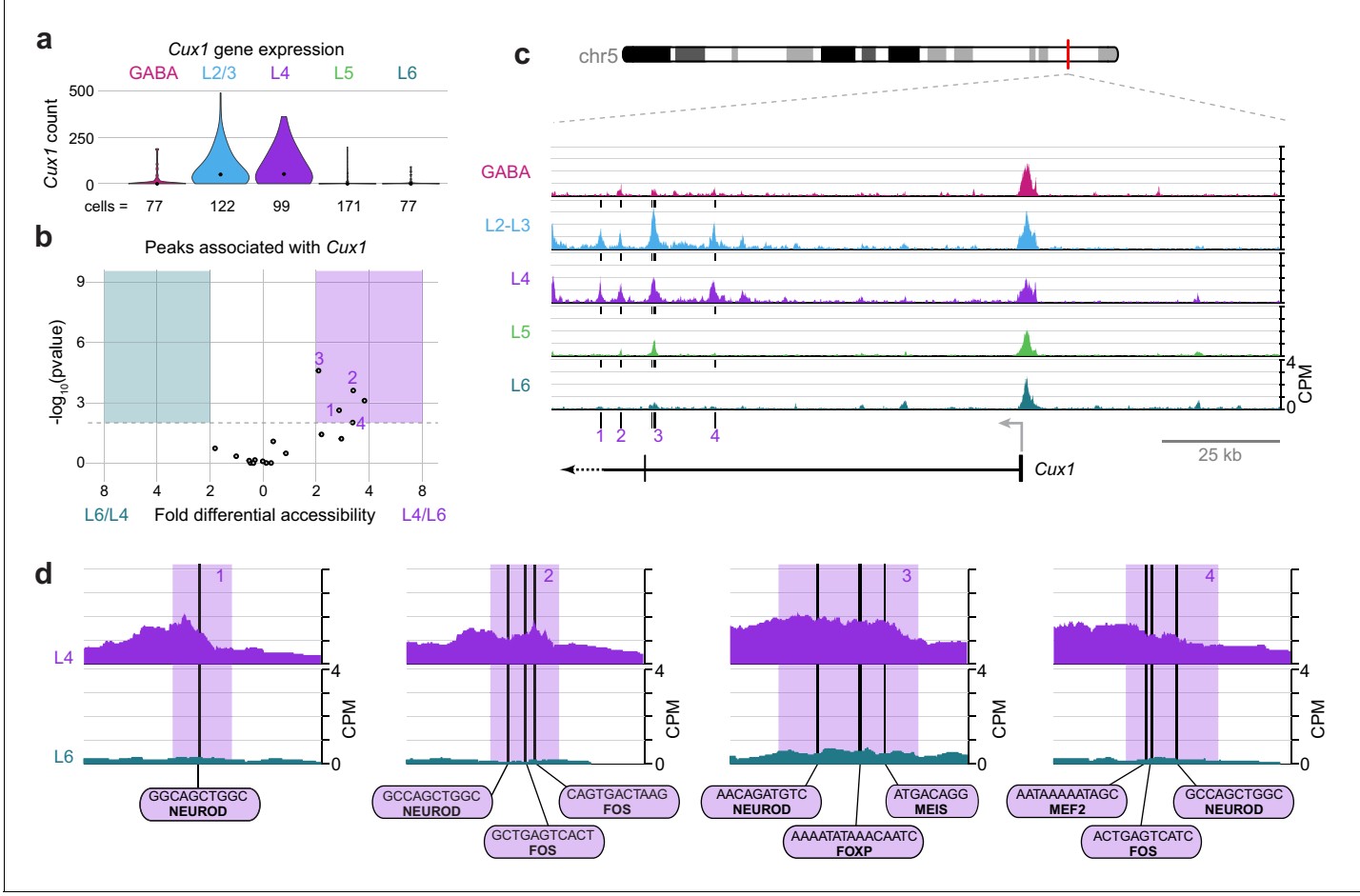

**Figure 10.** Cell class-specific regulatory domains downstream of the *Cux1* TSS. (a) *Cux1* gene expression distributions based on scRNA-seq in each neuronal cell class. (b) Volcano plot shoing all peaks that are positionally associated with *Cux1* in a pairwise comparison between L6 and L4 cell classes. Significantly differentially accessible peaks (adjusted p-values < 0.01 and > 2 -fold change in accessibility score) are highlighted as L6-specific (green box) or L4-specific (purple box). (c) Chromatin accessibility near the *Cux1* gene in each cell class (155 kb window; mm10 chr5:136,465,981–136,620,981). Vertical lines below the tracks represent the locations of peaks that are significantly more accessible in L4 compared to L6. (d) 1 kb windows centered on four numbered peaks that contain putative TF binding sites. TF motif locations are marked by black bars. CPM, counts of overlapping sequenced fragments at each position per million mapped reads for each cell class. All fragment overlap panels are plotted on the same scale (0 to 4 CPM).

The following source data is available for figure 10:

**Source data 1.** *Cux1* expression values used to generate the plot in *Figure 10A*.
**Source data 2.** Peak statistics for peaks positionally associated with *Cux1*, used to generate *Figure 10B*.

## Discussion

In the mouse visual cortex, at least 49 transcriptomic cell types have been defined by single cell RNA-seq (*Tasic et al., 2016*). As a first step in uncovering layer-specific regulatory networks, including regulatory transcription factors and their targets, we performed ATAC-seq on small populations of cells derived from layer-specific transgenic lines (*Figure 1*). Comparisons of our data to previously published ChIP-seq (*Figure 2*) and scRNA-seq (*Figures 4* and *5*) enabled identification of potentially significant regulatory sites. By restricting our analysis to a single cortical region, the mouse visual cortex, and through the use of layer-specific Cre lines, our data provide examination of the chromatin accessibility state at the resolution of layer-specific cortical cell classes (*Figures 9* and *10*). Previous studies have examined the state of cells across whole brain (*Yue et al., 2014*), in the whole

cerebellum (*Frank et al., 2015*), or in pooled GABAergic or glutamatergic cells across all layers in the entire mouse cortex (*Mo et al., 2015*). The cell class resolution of our study allowed us to identify open chromatin sites that are unique to each layer-specific cell class, and assign accessible sites from previous studies on more heterogeneous populations to more specific cell classes (*Figure 3—figure supplements 3* and *4*). In doing so, we were also able to build a network of transcription factor interactions that may be responsible for maintaining the identity of layer-specific cell classes (*Figure 7*).

Analysis of TF motif enrichment and accessible motif targets were used to identify putative regulatory TFs for cell classes from L2/3, L4, and L6 (*Figure 6*). However, distinct transcription factor families for L5 were not clearly identified by our analysis. This result may be due to the heterogeneity of cell types that are present in the L5 cell class defined by the *Rbp4-Cre* line. *Rbp4-Cre* labels at least nine transcriptomic cell types, including Layer 5a cells, which show some similarity to L4 cell types, and L5b types, which share some transcriptomic patterns with L6 cell types (*Tasic et al., 2016*). This heterogeneity may mask detection of distinct networks that regulate the cell types within L5, which could be uncovered in future studies that will require new Cre-driver lines or other labeling strategies to access L5 subtypes. We also note that L6 includes several transcriptomic cell types that were not surveyed in the current study: only corticothalamic L6a cells are labeled by the *Ntsr1-Cre* driver line (*Tasic et al., 2016*).

Development of laminar cell type identities occurs through several waves of differentiation driven by sequential changes in chromatin and transcriptional landscapes (*Telley et al., 2016*). In our analysis, we see largely distinct networks of interactions that define upper and lower layers of the cortex. Previous studies have shown that layer-specific populations arise at distinct times in development. Several of the TFs we predict to be key regulators of layer-specific transcription have been examined for their function in cortical development. Loss of *Tbr1*, which is a key lower-layer regulator expressed by lower-layer neurons, results in gross abnormalities in cortex. Curiously, in rostral cortical regions, *Tbr1* knockouts appear to lack L6, while other layers appear normal, whereas in caudal regions, *Tbr1* knockouts present a generally disorganized cortical structure (*Hevner et al., 2001*). These findings suggest that independent networks may be involved in upper versus lower laminar development in these regions. Double knockout of *Pou3f2* and *Pou3f1*, which we predict are key regulators in upper cortical layers, results in incorrect migration of upper cortical neurons (layers 2/3 and 4), but not lower layer neurons (*McEvilly et al., 2002*; *Sugitani et al., 2002*). Loss of key TFs can also affect layer-specific cortical projection patterns: knockdown of *Cux1* ablates ipsilateral cortico-cortical projections of upper-layer neurons (*Rodríguez-Tornos et al., 2016*), and loss of *Tbr1* results in truncated corticothalamic projections by L6 neurons (*Hevner et al., 2001*).

While these broad phenotypes show the importance of these factors in the development of cortical structure and function, specific interactions previously studied in development are frequently not found in our accessibility data from adult cortex. For example, a previously described regulatory interaction between RORB and its biding site near *Pou3f2*, which results in down-regulation of *Pou3f2* during laminar development of L2/3 and L4 (*Oishi et al., 2016*), is not part of our putative regulatory network (*Figure 7*). In single-cell RNA-seq of adult visual cortex, we often see simultaneous expression of *Rorb* and *Pou3f2* in Layer 2/3 and Layer 4 cells (*Tasic et al., 2016*, *Figure 6—figure supplement 2*), suggesting that this interaction may not be present in the adult visual cortex. We investigated this discrepancy between adult and developmental states by examining the RORB target sites that were previously described near the *Pou3f2* gene (*Figure 11*). We found very little chromatin accessibility at or around these RORB-binding sites, which suggests that the chromatin state in adult cells prevents this interaction, thus allowing coexpression of *Rorb* and *Pou3f2* after differentiation (*Figure 6—figure supplement 2*). Our network also lacks a previously reported interaction between POU3F2 and *Foxp2* through a highly conserved POU3F2-binding site in the *Foxp2* gene that is mutated in human compared to most other species (*Maricic et al., 2013*). In reporter assays, the conserved POU3F2 motif, which is present in mouse, acts as a cis-regulatory enhancer element (*Maricic et al., 2013*). In adult visual cortex, the chromatin is inaccessible near this POU3F2 motif (*Figure 11*), suggesting that POU3F2 does not bind to this site to regulate *Foxp2* expression. This is consistent with gene expression data from adult cortex, where we see no expression of *Foxp2* in cell classes that express *Pou3f2* (layers 2–5, *Figure 6—figure supplement 2*). Further chromatin accessibility experiments would be needed to determine if this regulatory element is accessible during development or other cell types in visual cortex, or in other brain regions.

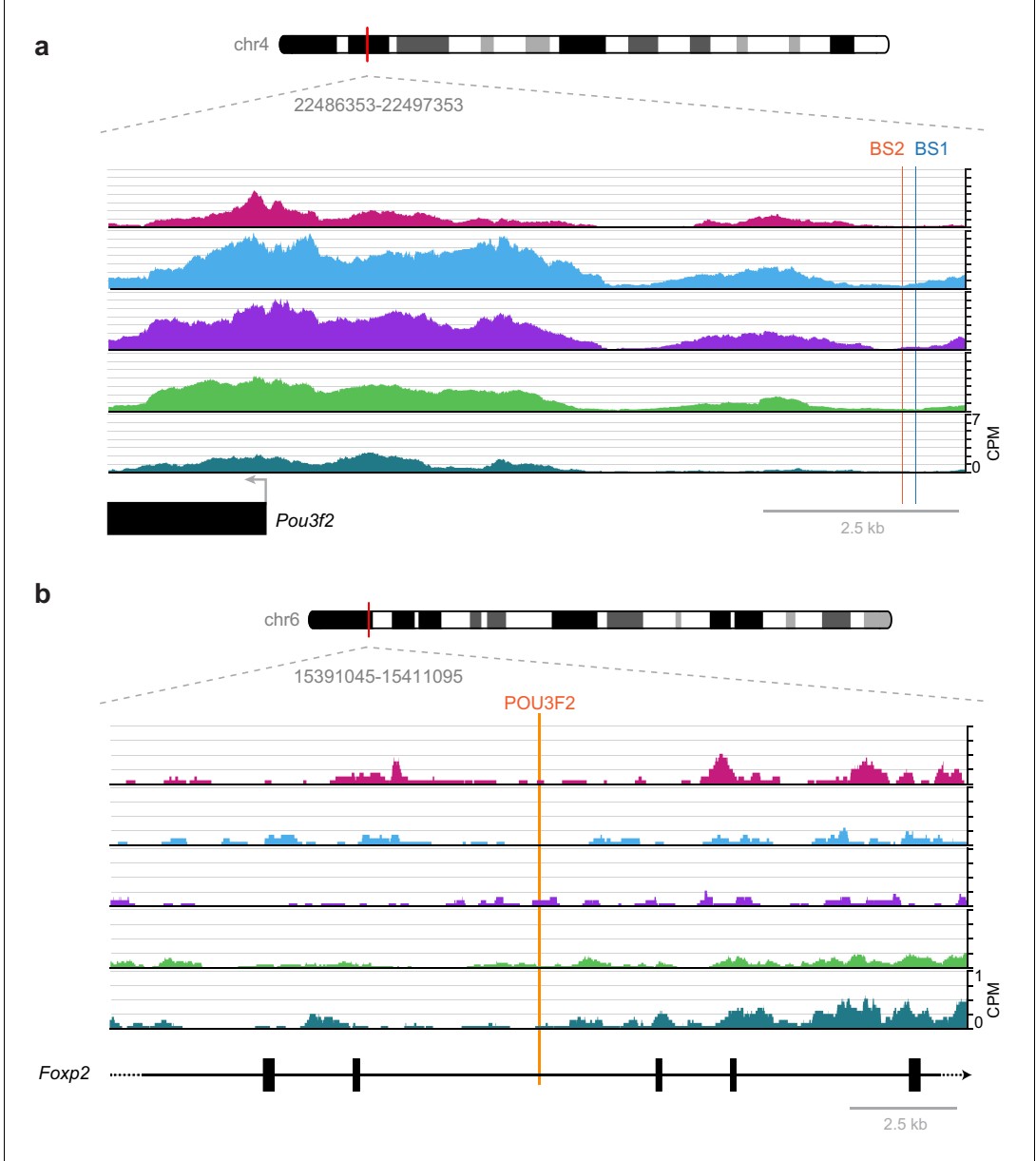

**Figure 11.** Previously published TF-binding sites near *Pou3f2* and *Foxp2* observed during development are not accessible in adult mouse cortex. (a) Previously described RORB-binding sites (BS1 and BS2) near the *Pou3f2* TSS are not accessible in any of the adult cell classes we examined. (b) Same as in (a), but for a POU3F2 binding site in the *Foxp2* gene. CPM, counts of overlapping sequenced fragments at each position per million mapped reads for each cell class.

A recent publication has shown that FOXP2 directly suppresses *Mef2c* expression in the striatum, where knockout of *Foxp2* allows higher *Mef2c* expression, which, in turn, suppresses synaptogenesis (*Chen et al., 2016*). Our network analysis suggests that FOXP2 binding may also repress *Mef2c* in lower layers of the adult visual cortex (*Figure 7*). We examined the FOXP motifs near *Mef2c* exon three that were identified in this recent study (*Figure 12B*). In adult cortex, we observe only a modest and statistically insignificant decrease in their accessibility in lower layers. Thus, we expanded our search to all FOXP motif-containing peaks near *Mef2c* in the cortex (*Figure 12A*). We found two peaks upstream of the *Mef2c* TSS that are less accessible in L6 than in upper layers. Therefore, FOXP2 may regulate *Mef2c* by binding different sites in these different tissues: sites upstream of TSS in the cortex, and sites near exon three in the striatum. This hypothesis could be tested by

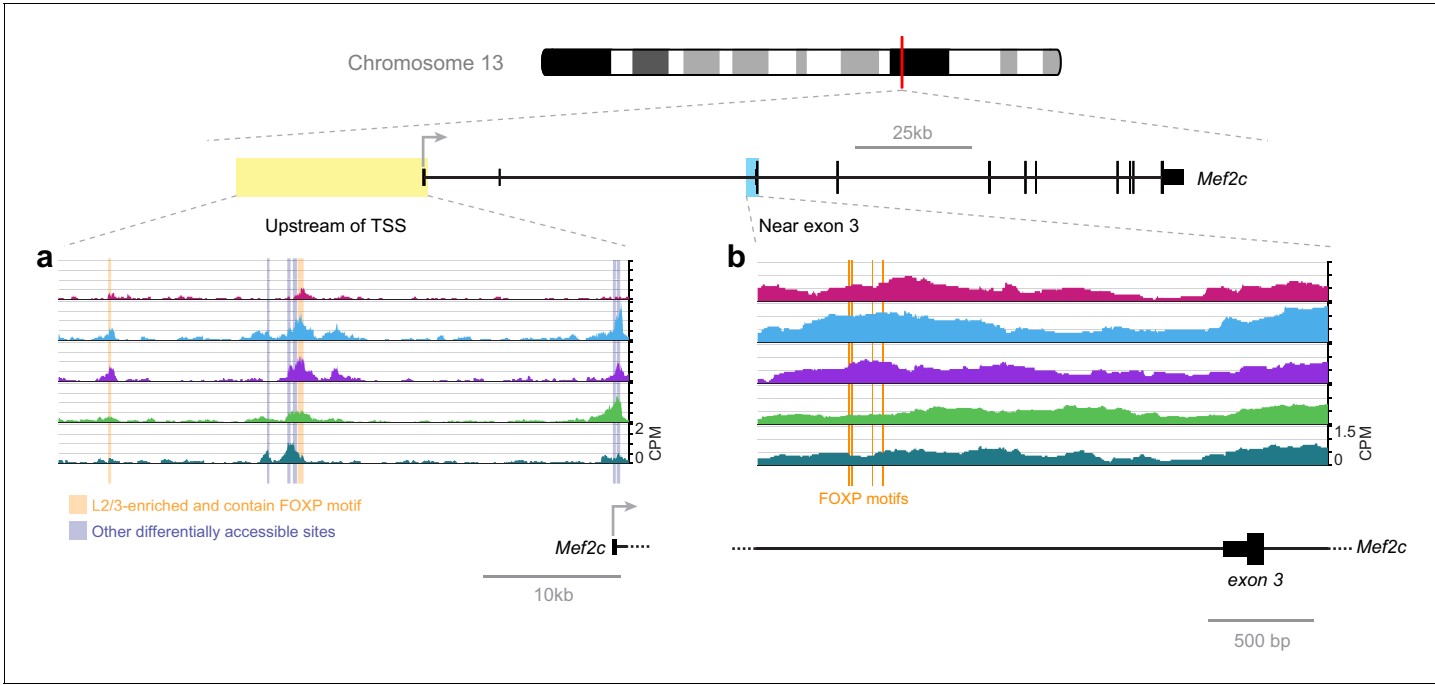

**Figure 12.** FOXP motif accessibility within or near the *Mef2c* gene. (a) Chromatin accessibility upstream of the *Mef2c* TSS. Orange boxes, putative FOXP binding sites that are significantly less accessible in L6 than in upper layers; Gray boxes, differentially-accessible sites that do not contain FOXP motifs. (b) FOXP motifs identified as direct targets of FOXP2 in a previous study (*Chen et al., 2016*). CPM, counts of overlapping sequenced fragments at each position per million mapped reads for each cell class.

future experiments that measure the chromatin accessibility landscape of the striatum, and genome-wide interaction sites of FOXP2 by ChIP-seq in striatum and cortex, as well as the effect of decreasing FOXP2 levels through cell-type specific knockout or knockdown on these chromatin landscapes.

These comparisons emphasize the importance of developmental, anatomical, and cell type context in studies of chromatin accessibility and transcriptomic regulation. We note that several transcription factors identified in our network are known to be expressed in development. Some of their target sites in development may be retained in adulthood, while others may be different due to changes in accessibility. Future studies performed on layer-specific cell classes in development will be needed to determine how changes to the epigenetic landscape affect transcription during development.

Our sampling strategy utilized only Cre lines that yielded at least 500 tdTomato-positive FACS-isolated cells per animal. This limited our genetic access to relatively abundant cell classes in the visual cortex, all of which label multiple cell types that can be distinguished by single-cell RNA-seq (*Figure 1C*). Because GABAergic cell types are much less abundant in the neocortex than glutamatergic cells, we sampled only a single, pan-GABAergic cell line, *Gad2-IRES-Cre*. A previous study was able to examine *Pvalb+* and *Vip+* (but not *Sst+* or *Ndnf+*) GABAergic cell type classes using the whole mouse cortex (*Mo et al., 2015*) by utilizing the INTACT nuclear labelling technique. The higher spatial resolution of our glutamatergic data, but lower cell-type resolution among GABAergic types makes our study complementary to this work. Further advances in low-input chromatin accessibility assays, low input ChIP-seq methods, and improvements in Cre-driver specificity will increase our ability to analyze the epigenomic state of specific neural cell types.

Our integrated analysis of chromatin accessibility, gene expression, and TF motif enrichment provides an unprecedented look at the key interactions that underlie layer-specific transcriptomes. We were able to identify layer-wide patterns of TF accessibility from analysis of the entire set of peaks from each layer, and found interactions that may drive expression of key transcriptomic regulators *Nfia* and *Cux1* by looking at coordinated TF expression and motif accessibility. Further research into the roles that these and other TFs play in development and maintenance of layer-specific cell classes

will enhance our understanding of the functional consequences of these transcriptomic programs. Conditional loss-of-function experiments in specific cell classes at specific times in development or adulthood would help to disentangle the specific roles from more global roles of these TFs. For instance, constitutive, homozygous loss-of-function of *Foxp2* is lethal by postnatal day 21, and causes developmental delays and motor dysfunction that may interfere with studies of adult brain function (*French et al., 2007*), whereas heterozygous *Foxp2* knockout mice display few differences in cortical structure (*Fujita et al., 2008*). Our network suggests that *Foxp2* could be a key regulator of L6-specific transcriptional program in VISp. Investigation of this role of Foxp2 would require generation of a conditional knockout in specific cortical cell populations, and perhaps at specific times in development or adulthood (for example, through conditional alleles combined with Cre/CreER lines). Another area that would greatly benefit from further study is assessing the correspondence between chromatin modifications and chromatin accessibility. New techniques for ChIP-seq enable enrichment of modified histones from as few as 500 cells (*Lara-Astiaso et al., 2014*), which may allow the correspondence between accessibility and chromatin modifications in specific cell classes to be studied directly. The synthesis of these data modalities will further our understading of the mechanisms of cell type-specific transcriptional regulation. This new knowledge will be useful for building new cell-type specific genetic tools, and for testing the reprogramming potential of layer-specific cell classes (*De la Rossa et al., 2013*).

## Materials and methods

### Mouse breeding and husbandry

All mice were housed at the Allen Institute for Brain Science under Institutional Care and Use Committee protocols 0703, 1208, and 1508. No more than five animals per cage were maintained on a regular 12 hr day/night cycle, with water and food provided *ad libitum*. Animals were maintained on the C57BL/6J background, as described previously (*Tasic et al., 2016*). We used only heterozygous animals that were positive for both Cre-recombinase drivers and tdTomato reporter transgenes (we used only the *Ai14* Cre reporter transgene, (*Madisen et al., 2010*). Parental mouse strains used: *Ai14* (RRID:IMSR_JAX:007914), *Cux2-CreERT2* (RRID:MMRRC_032779-MU), *Gad2-IRES-Cre* (RRID: IMSR_JAX:010802), *Ntsr1-Cre_GN220* (RRID:MMRRC_030648-UCD), *Rbp4-Cre_KL100* (RRID: MMRRC_031125-UCD), and *Scnn1a-Tg3-Cre* (RRID:MGI:3850203). *Cux2-CreERT2* mice were treated with tamoxifen using a single dose of 40 µL of 50 mg/mL tamoxifen dissolved in corn oil and administered by oral gavage at postnatal day (P)10–14. Animals with anophthalmia or microphthalmia were excluded from experiments. Expression patterns for the five Cre lines used in this study were previously characterized as part of the Allen Institute Connectivity Atlas Transgenic Characterization pipeline (*Harris et al., 2014*). These results are openly available online at http://connectivity.brain-map.org/transgenic.

### Isolation of 500 cell populations

We generated single-cell suspensions of fluorescently labeled neurons as described previously (*Tasic et al., 2016*). Briefly, we used an isoflurane chamber to anesthetize adult male mice (P56 ± 3), decapitated them, removed the brains, and transferred them immediately to freshly prepared, ice-cold artificial cerebrospinal fluid (ACSF: 126 mM NaCl, 20 mM NaHCO3, 20 mM dextrose, 3 mM KCl, 1.25 mM NaH2PO4, 2 mM CaCl2, 2 mM mgCL2, 50 µM DL-AP5 sodium salt, 20 µM DNQX, and 0.1 µM tetrodotoxin, mixed then bubbled with 95% $O_2$/5% $CO_2$ carbogen gas). We sectioned the brains to generate 400 µm–thick sections using a Leica VT1000S vibratome with a chilled chamber, and immediately transferred the slices to a bubbled ACSF-containing chamber at room temperature. Individual slices of interest were microdissected in a Petri dish while submerged in ACSF under a fluorescence dissecting microscope. Dissected tissue was transferred to a microcentrifuge tube containing ACSF with 1 mg/mL pronase (Sigma, Cat#P6911) for 70 min at room temperature. After incubation, with tissue pieces settled at the bottom of the tubes, ACSF with pronase was exchanged twice with ACSF containing 1% fetal bovine serum (FBS). We next dissociated the tissue into a single-cell suspension by trituration through Pasteur pipettes with polished openings of 600 µm-, 300 µm-, and 150 µm-diameter. 500 cells were sorted into a well of an 8-well PCR strip containing 2 µL ACSF on a BD FACSAriaII SORP with a 130 µm nozzle at 10 psi sheath pressure, and in the

single-cell sorting mode. We excluded dead cells by labelling with DAPI (DAPI*2HCl, Life Technologies Cat#D1306) added to the single-cell suspension at 2 ng/mL. We retained only cells that had high tdTomato fluorescence and low DAPI labeling.

For mES cell populations, G4 ES cells (RRID:CVCL_E222, Lunenfeld-Tanenbaum Research Institute, Mount Sinai Hospital) grown on mouse embryonic fibroblast (MEF) feeders were passaged onto gelatin plates to dilute/remove feeder cells, and they were made into single-cell suspensions by treatment with Trypsin-EDTA (Thermo Fisher Scientific, Cat#25300054). Cells were washed twice in PBS containing 1% FBS, and were then resuspended in PBS containing 1% FBS and 2 ng/mL DAPI. 500-cell populations of DAPI-negative ES cells were sorted into individual wells of an 8-well PCR strip containing 2 μL of PBS.

In total, we used 23 animals, with at least two animals per Cre line, which yielded 32 samples of 500 cells each. Of these, 25 libraries were successfully amplified, 17 libraries passed sequencing quality control checks after MiSeq (Illumina), and 14 of these 17 were sent for sequencing on HiSeq (Illumina). One library (Rbp4 sample 3) was sequenced using an entire flow-cell on a MiSeq instead of HiSeq. The HiSeq samples and the Rbp4 sample 3 were used for downstream analysis.

## ATAC-seq of cortical cell populations and mES cells

For low-input ATAC-seq, we utilized a previously published protocol (*Lara-Astiaso et al., 2014*). Immediately after cell collection by FACS, cells were lysed with 25 μL Lysis Buffer (10 mM Tris-HCl pH 7.4, 10 mM NaCl, 3 mM MgCl2, and 0.1% IGEPAL CA-630). Nuclei were pelleted by centrifugation at 500 x g for 30 min at 4°C in a refrigerated microcentrifuge. After pelleting, the supernatant was removed and nuclei were resuspended by repeated pipetting in 25 μL Reaction Mix (12.5 μL Nextera TD Buffer, 2 μL Nextera TD Enzyme, and 10.5 μL water). Samples were then tagmented by incubation at 37°C for 1 hr in a heat block. After tagmentation, the reactions were stopped with addition of 5 μL Cleanup Buffer (900 mM NaCl, 300 mM EDTA), 2 μL 5% SDS, and 2 μL Proteinase K and incubation at 40°C for 30 min. Tagmented DNA was purified using AMPure XP beads (Beckman Coulter) at a ratio of 1.8:1 beads to reaction volume, with a final resuspension in 11 μL TE. For indexing and amplification, we added 15 μL KAPA HotStart Ready mix (Kapa Biosystems, Cat# KK2602) and 2 μL each of Nextera i5 and i7 indexed amplification primers (Illumina). These samples were incubated at 72°C for 3 min, then PCR amplified as follows: 95°C for 3 min; 9 cycles of 98°C for 20 s, 65°C for 15 s, and 72°C for 15 s; final extension 72°C for 1 min. Samples were then purified using AMPure XP as above, reamplified for nine additional cycles under the same conditions, and then purified once more as before using Ampure XP beads to produce 11 μL final volume. Library quality and quantity were assessed using 1 μL of the final, purified DNA on a BioAnalyzer High Sensitivity DNA chip (Agilent Technologies).

## Sequence alignment and peak analysis

High-quality libraries were sequenced on an Illumina HiSeq or MiSeq to obtain 50 bp paired-end reads. Sequencing results are available in GEO with accession number GSE87548. Reads in FASTQ format were aligned to GRCm38 (mm10) using Bowtie v1.1.0 (RRID:SCR_005476) (*Langmead et al., 2009*) with settings –m 1 –X 2000 –chunkmbs 256 in paired-end mode. Unaligned reads were processed with the Trim galore wrapper for Cutadapt (RRID:SCR_011841) (*Martin, 2011*) to remove Nextera primer sequences (settings: –nextera –paired –three_prime_clip_R1 1 –three_prime_clip_R2 1), and were then aligned again using Bowtie (settings –m 1 –X 2000 −3 1 –chunkmbs 256). The Samtools collection (RRID:SCR_002105) (*Li et al., 2009*) was used to sort, remove PCR duplicates (rmdup), index BAM files (index), and calculate library statistics (flagstat). We used CollectInsertSizeMetrics.jar from Picard v1.110 (RRID:SCR_006525) (*BroadInstitute, 2015*) to analyze fragment size statistics, and preseq v0.1.0 (*Daley and Smith, 2013*) to analyze library sequencing saturation. The CENTIPEDE package for R (*Pique-Regi et al., 2011*) was used to analyze insertions near ATF2 motif locations obtained from the SwissRegulon database (RRID:SCR_005333) (*Pachkov et al., 2013, 2007*). To downsample BAM files, the data were sorted by name instead of location using Samtools sort in SAM format, then the R sample function was used to select random read pairs without replacement, and a custom Perl script filtered the selected reads. Samtools view was then used to convert files back to BAM format. After downsampling, aligned reads were analyzed for peak and region enrichment of open chromatin using HotSpot v4.1 (*John et al., 2011*) with default settings.

For differential binding analysis and to cluster samples based on ATAC-seq enrichment, we used DiffBind v1.16.3 (RRID:SCR_012918) (*Ross-Innes et al., 2012*; *Stark and Brown, 2011*) for overlap analysis (dba) and weighted overlap analysis (dba.count). DESeq2 v1.10.1 (*Love et al., 2014*) was used for contrast analysis between pairwise groups of Cre line samples using DiffBind functions (dba.analyze) with settings to use DESeq2 (method = DBA_DESEQ2). DiffBind was also used to calculate merged peak locations, based on the outer boundaries of overlapping peaks from all cell lines analyzed, and peak accessibility scores for each replicate were calculated using DiffBind with the default setting DBA_SCORE_TMM_MINUS_EFFECTIVE, which uses trimmed mean of M-values (TMM) normalization (*Robinson and Oshlack, 2010*) built into the edgeR package (*Robinson et al., 2010*) using read counts minus control (genomic) read counts and full library size. Peak TMM scores for each cell class and all DiffBind pairwise comparison p-values are reported in *Supplementary file 1B*. Clustering of raw peak data was performed using the R function hclust and the 'complete' method using DiffBind scores from neural cell types from the 7,500 most highly differentially accessible peaks among neural cell types, as ranked by adjusted p-values.

## Comparisons to previously published chromatin accessibility and ChIP-seq data

For comparisons between our datasets and whole cortex INTACT-ATAC-seq from *Camk2a-Cre*, *Pvalb-Cre*, and *Vip-Cre* labeled cells (*Mo et al., 2015*), we retrieved paired-end, raw read data from GEO Series GSE63137 in FASTQ format. These datasets were aligned and downsampled to 3.2 M read pairs as described above, and HotSpot was used to call peaks as for our samples. For comparisons to ChIP-seq results from Camk2a, we downloaded SICER peak calls from GEO Series GSE63137 in BED format. For comparison to ENCODE DNase-seq data, we downloaded HotSpot peak datasets in BED format from the Mouse ENCODE data portal (www.mouseencode.org/data), and selected up to three replicates for each adult tissue available, as well as for ES-E14 cells. Whole-dataset comparisons were performed using DiffBind with peak overlaps only (dba; no weighted overlap analysis). Overlap frequency counts were calculated using the GenomicRanges R package (*Lawrence et al., 2013*), after pooling and using GenomicRanges to merge overlapping peaks among all replicates for ENCODE Whole Brain DNase-seq peaks, *Camk2a-Cre* ATAC-seq peaks, or combined ATAC-seq *Pvalb-IRES-Cre* and *Vip-IRES-Cre* peak datasets.

## scRNA-seq datasets and peak assigments to genes

To compare chromatin accessibility with gene expression, we used single cell RNA-seq data from a previous study (*Tasic et al., 2016*; GEO accession GSE71585). In that study, we characterized transcriptomes of cells isolated from the same Cre lines used in this study. To generate 'average' gene expression patterns for cells from these Cre lines (i.e., cell-class transcriptomic average), we downsampled RNA-seq data for each cell of interest to 1 million mapped reads (downsampled data are available in *Supplementary file 3*). In total 546 tdTomato+ cells were used to generate these cell-class transcriptomic averages: 77 *Gad2-IRES-Cre;Ai14* cells, 122 *Cux2-CreERT2;Ai14* cells, 99 *Scnn1a-Tg3-Cre;Ai14* cells, 171 *Rbp4-Cre;Ai14* cells, and 77 *Ntsr1-Cre;Ai14* cells. To define differentially expressed genes between each pair of cell classes, we used DESeq2 (*Love et al., 2014*). To assign each peak to the nearest gene, we applied the *nearest* function from GenomicRanges in R to the merged peak set from DiffBind and RefSeq TSS gene annotations retrieved from the UCSC Genome Browser database (RRID: SCR_005780). These peak-gene associations are summarized in *Supplementary file 1C*. For downstream analysis, replicate values for ATAC-seq data from each Cre line were averaged. We then calculated the correlation between peak accessibility and associated gene transcription using the sample Pearson correlation coefficient. To determine if average peak accessibility and average gene expression for any gene-peak pair were associated more strongly than expected by chance, we compared these correlations to 10 randomly-permuted datasets (*Figure 4—figure supplement 2*).

## Pairwise comparisons of peak accessibility and gene expression

To compare pairwise peak accessibility to pairwise gene expression data (*Figure 4A*), we first calculated differentially-expressed genes using DESeq2 for each pair of neuronal cell classes. For each pairwise comparison, we filtered the genes to select those that were significantly

differentially expressed between the two cell classes (adj. p-value < 0.001), and assigned them into one of the two groups based on the cell class with higher average expression for that gene. For each of these two groups, we then selected all peaks from the two cell classes involved in the comparison that were associated with the differentially-expressed genes. We then determined if the peak accessibility values for the two classes were significantly different using a Mann-Whitney *U* test. We adjusted the results for multiple comparisons using Bonferroni correction.

## ATAC-seq and RNA-seq module analysis

We assigned peaks and genes to modules using a two-step *k*-means clustering process. This was done to first identify common patterns of accessibility and expression, then to use those patterns to build modules found both among ATAC-seq peaks and gene expression. We first selected the peaks for clustering as those which were significantly differentially accessible in at least one comparison between pairs of glutamatergic cell classes (adjusted DiffBind p-value < 0.01) and had at least a 4-fold difference in average TMM accessibility score. We then scaled peak accessibility scores between 0 and 1 for each peak by subtracting the minimum value for each peak across glutamatergic classes, and then dividing by the maximum. For building gene expression modules, we selected only differentially expressed genes and performed the same scaling as for peaks (DESeq2 pval < 0.05, fold change > 2). These sets of peaks and genes were then clustered separately using *k*-means clustering to build an initial set of patterns found in the chromatin accessibility and gene expression data (*Figure 5—figure supplement 1*). The vectors used for clustering were scaled average peak accessibility values (for peak module analysis) and scaled average gene expression values (for gene module analysis), described above. These vectors each have four dimensions, one for each glutamatergic cell class. Because unsupervised *k*-means clustering is dependent on random selection of initial cluster centroids, we manually selected a set of 8 clusters based on the initial *k*-means cluster results as seeds for a second round of clustering to obtain a convergent set of modules. We chose four patterns based on selective, high accessibility/expression for each of the four glutamatergic cell classes, as well as four other patterns that were frequently observed in both peaks and genes (Upper+, high in L2/3 and L4; Lower+, high in L5 and L6; L4-, low in L4 only; and L6-, low in L6 only). To use these patterns as seed centroids, we generated binarized centroid vectors. For each dimension, values > 0.5 were set to 1, and < 0.5 were changed to 0. We then ran *k*-means clustering on the selected peaks and genes using these eight cluster centers as seed values to generate final peak and gene modules. We tested the significance of peak-gene module associations by counting the frequency with which peaks in each peak module were positionally associated with genes in each gene module, then calculated enrichment or depletion using Fisher's exact test. To account for multiple comparisons, we used Benjamini and Hochberg correction.

## Motif enrichment analysis

To calculate enrichment of motifs in our peak modules (defined above), we first chose a background set for each module. The background sets were selected by choosing all modules whose *k*-means cluster centers were < 0.5 for any cell classes in which the foreground module had *k*-means cluster centers > 0.5. This selection ensured that peaks in the foreground module and background module did not share the same accessibility pattern in any of the glutamatergic cell classes (*Figure 6—figure supplement 1*). We next removed any peaks with a width > 400 bp, and retrieved the sequences corresponding to each peak in the foreground and background sets. Files containing these sequences were submitted to Analysis of Motif Enrichment (AME v4.10.1, *McLeay and Bailey, 2010*), part of the MEME suite (RRID:SCR_001783), to calculate enrichment of sequence motifs in each of the eight peak modules compared to peaks in the dissimilar background sets. For AME analysis, we used the JASPAR 2016 VERTEBRATES motif database (RRID:SCR_003030) (*Mathelier et al., 2016*), with the addition of NEUROD2 motifs which have been previously identified, but were not included in this database (*Fong et al., 2015*). We examined the AME results for each module, and compiled a set of motifs that were found to be highly significantly associated with differential peak accessibility: CUX, DLX, etc. To find depleted motifs, AME was also performed with foreground and background peak sets reversed. AME results for each peak module are available in *Supplementary file 2A*. For downstream analysis of peaks that contain these TF target motifs, we searched all peaks for motif occurrences using Find Individual Motif Occurrences (FIMO, *Grant et al., 2011*). Selected

FIMO results, with the locations of each differentially accessible motif among our peak sets, can be found in *Supplementary file 2B*. To identify the cognate TFs that may bind these motifs, we used a single exemplar of these TF families to search the TreeFam database (RRID:SCR_013401) to identify related transcription factors (*Ruan et al., 2008*), then filtered these factors for those with an average gene expression count > 5 for at least one glutamatergic cell class. We then performed literature searches for each remaining factors to determine if they are known to not bind to the motifs identified in our AME results, and excluded those factors. To footprint TF occupancy, we used the locations of motif families identified by FIMO within our peak set, and plotted the position of the 5' end of the sequenced fragments in a single replicate of BAM-formatted mapped reads for each cell class as the Tn5 insertion sites for that cell class. These footprints were then normalized to the millions of reads in each BAM file. Motif LOGOs were generated using WebLogo v3.5.0 (*Crooks et al., 2004*).

## Network analysis

To build a regulatory network, we selected regulatory source nodes using the flowchart in *Figure 7—figure supplement 1A*. We started with differentially enriched motifs from AME analysis, and resulting TreeFam TF candidates, described above. We then assigned a gene module to each TF, by calculating the Pearson correlation between the average gene expression and the $k$-means centers of each gene module. The module with the highest score was assigned to the gene. We then looked at literature to determine if the gene product was a likely repressor or activator of transcription. If the gene product was likely to be an activator, we checked to see if the TF target motif was enriched among peaks in the peak module that matched the assigned gene module. If the gene product was likely to be a repressor, we checked for depletion of the TF target motif in the corresponding gene module. For each set of motifs and gene modules, we then selected the highest-expressed gene that passed the above criteria as a putative activator or repressor source node.

We next selected edges (TF motifs) and targets (TF genes) using the flowchart provided in *Fig. 7-figure supplement 1BFigure 7—figure supplement 1B*. For each differentially enriched motif selected from our AME analysis, we used FIMO to find all instances of the motif in the sequences of our peaks. We then filtered motifs to obtain only those that were significantly associated with at least one peak module (Fisher's exact test p-value < 0.01), then filtered for motifs that were found in peaks that were positively associated with a gene expression module (Fisher's exact test p-value < 0.01). We then selected motifs that were found in peaks that were significantly differentially accessible in at least one pairwise comparison between glutamatergic cell classes (DiffBind p < 0.001), and were associated with a differentially expressed gene (DESeq2 p < $1\times10^{-5}$). To plot the resulting network, we restricted our target set to motifs in peaks that were positionally associated with known TF genes listed in the AnimalTFDB database (RRID:SCR_001624) (*Zhang et al., 2012*). Finally, we grouped our filtered motifs based on whether the TF that binds the target motif is known to be a repressor or an activator. For activators, we did a final filtering step to select only motifs in peaks whose accessibility was positively correlated with expression of the activator gene (Pearson correlation coefficient > 0.3). For repressors, we selected motifs in peaks that were negatively correlated with expression of their putative regulator (Pearson correlation coefficient < −0.3). The final table of filtered associations is available as *Supplementary file 2C*. Finally, we plotted the resulting network diagram using Cytoscape v.3.4.0 (*Shannon et al., 2003*).

## Data analysis and presentation tools

Source code for our analyses are available online at: https://github.com/hypercompetent/Gray2017eLife. Analysis and visualization of the data presented in this paper were made possible through the use of several R packages: dplyr (*Wickham and Francois, 2016*), gdata (*Warnes et al., 2015*), matrixStats (*Bengtsson, 2016*), purrr (*Wickham, 2016*), and reshape2 (*Wickham, 2007*) for general manipulation of datasets; ggplot2 for plotting of most graphs and figures (*Wickham, 2009*); gplots for plotting of some heatmaps (*Warnes et al., 2016*); viridis (*Garnier, 2016*) and RColor-Brewer (*Neuwirth, 2014*) for heatmap color selection; dendextend for plotting dendrograms (*Galili, 2015*); Gviz for plots of chromosome ideograms (*Hahne and Ivanek, 2016*); UpSetR for plotting overlaps between sets of ATAC-seq peaks (*Lex et al., 2014*); GenomicRanges for calculations involving peak and TSS positions in the genome (*Lawrence et al., 2013*); rtracklayer (*Lawrence et al.,*

*2009*) for access to RefSeq data from the UCSC Genome Browser database; GenomicAlignments for manipulating data from paired-end BAM files (*Lawrence et al., 2013*); DESeq2 for calculations of differential gene expression and differential peak accessibility (*Love et al., 2014*); DiffBind for comparisons between ATAC-seq samples (*Stark and Brown, 2011*); and BSgenome for genomic sequence retrieval (*Pagès, 2016Pagès, 2016*). We have listed only packages that were invoked directly, though each package may require additional dependencies.

## Acknowledgements

We would like to thank Nadiya Shapovalova, Susan Bort, and Boaz Levi for assistance with FACS; Vilas Menon for discussions on data analysis; the Department of In Vivo Sciences at the Allen Institute for Brain Science for mouse colony management; and Boaz Levi and Trygve Bakken for critical reading of our manuscript. We wish to thank the Allen Institute founders, Paul G Allen and Jody Allen, for their vision, encouragement, and support.

## Additional information

### Funding

| Funder | Grant reference number | Author |
| --- | --- | --- |
| National Institute on Drug Abuse | 1R01DA036909-01 | Lucas T Gray Hongkui Zeng Bosiljka Tasic |

The funders had no role in study design, data collection and interpretation, or the decision to submit the work for publication.

### Author contributions

LTG, Conceptualization, Data curation, Software, Formal analysis, Investigation, Visualization, Methodology, Writing—original draft, Writing—review and editing; ZY, Software, Formal analysis, Visualization, Writing—review and editing; TNN, TKK, Investigation, Writing—review and editing; HZ, Supervision, Funding acquisition, Project administration, Writing—review and editing; BT, Conceptualization, Supervision, Funding acquisition, Writing—original draft, Project administration, Writing—review and editing

### Author ORCIDs

Lucas T Gray, http://orcid.org/0000-0002-8814-6818

Zizhen Yao, http://orcid.org/0000-0002-1210-4195

Thuc Nghi Nguyen, http://orcid.org/0000-0002-6466-5883

Tae Kyung Kim, http://orcid.org/0000-0001-9646-5969

Hongkui Zeng, http://orcid.org/0000-0002-0326-5878

Bosiljka Tasic, http://orcid.org/0000-0002-6861-4506

### Ethics

Animal experimentation: All mice were housed at the Allen Institute for Brain Science under Institutional Care and Use Committee protocols 0703, 1208, and 1508. No more than 5 animals per cage were maintained on a regular 12-h day/night cycle, with water and food provided ad libitum. Animals were sacrificed while under total aneshtesia induced by isofluorane to minimize suffering.

## Additional files

### Supplementary files

• Supplementary file 1. Libraries, Peaks, and associated statistics. (a) Library and alignment statistics. (b) Merged DiffBind peak locations, mean TMM scores per cell class, and pairwise DiffBind p-values. (c) Associations between peaks and nearest genes, average expression scores for associated genes, and pairwise DESeq2 differential expression p-values.

• Supplementary file 2. Motif analysis results. (a) Peak module AME results. (b) FIMO motif search results. (c) Network interaction table.

• Supplementary file 3. Downsampled RNA-seq data.

### Major datasets

The following dataset was generated:

| Author(s) | Year | Dataset title | Dataset URL | Database, license, and accessibility information |
|---|---|---|---|---|
| Gray LT, Yao Z, Nguyen TN, Kim SK, Zeng H, Tasic B | 2017 | Layer-specific ATAC-seq of the neurons of adult mouse visual cortex defined by Cre-driver lines | https://www.ncbi.nlm.nih.gov/geo/query/acc.cgi?acc=GSE87548 | Publicly available at the NCBI Gene Expression Omnibus (accession no: GSE87548) |

The following previously published datasets were used:

| Author(s) | Year | Dataset title | Dataset URL | Database, license, and accessibility information |
|---|---|---|---|---|
| Mo A, Mukamel EA, Davis FP, Luo C, Eddy SR, Ecker JR, Nathans J | 2015 | Epigenomic Signatures of Neuronal Diversity in the Mammalian Brain | https://www.ncbi.nlm.nih.gov/geo/query/acc.cgi?acc=GSE63137 | Publicly available at the NCBI Gene Expression Omnibus (accession no: GSE63137) |
| Tasic B, Menon V, Nguyen TN, Kim TK, Yao Z, Gray LT, Hawrylycz M, Koch C, Zeng H | 2016 | Adult mouse cortical cell taxonomy by single cell transcriptomics | https://www.ncbi.nlm.nih.gov/geo/query/acc.cgi?acc=GSE71585 | Publicly available at the NCBI Gene Expression Omnibus (accession no: GSE71585) |

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
