## [Decision Letter]

Thank you for submitting your article "Layer-specific chromatin accessibility landscapes reveal regulatory networks in adult mouse visual cortex" for consideration by *eLife*. Your article has been favorably evaluated by a Senior Editor and three reviewers, one of whom is a member of our Board of Reviewing Editors. The reviewers have opted to remain anonymous.

The reviewers have discussed the reviews with one another and the Reviewing Editor has drafted this decision to help you prepare a revised submission.

Summary:

The authors use ATAC-seq to probe chromatin accessibility profiles in four layer-specific classes of mouse visual cortical neurons, to identify regulatory elements that maintain layer-specific neuron identity. They present careful descriptive analyses of the data, combined with previous Allen Institute single-cell RNA-seq data from the same cortical region, and compared against relevant histone modification and RNA-seq data from other previously published work. Having identified a likely set of candidate regulatory regions, they present plausible guesses at the transcriptional factor regulatory network that maintains layer-specific adult cortical cell types, such as *Foxp2* being expressed in lower layers and repressing TFs for upper layers, and *Cux1* doing the reverse.

The referees unanimously agreed that as a data resource, the work is of extraordinarily high rigor and quality, and likely to be a treasure trove for future studies. The referees also agreed that the transcriptional network proposed in the latter half of the paper was speculative and lacked experimental validation. After discussion, we agreed that the paper is a strong contribution as a data resource, suitable for publication in *eLife*, and that rather than asking for additional experimental validation, various revisions could be made to strengthen the biological interpretation and significance of the data. We also propose some necessary revisions for clarity and for improving access to the data resource.

Essential revisions:

1) The lack of experimental validation of any of the predicted roles for transcription factors and their sites is problematic. Are there published data, such as relevant loss-of-function phenotypes, that you can interpret in light of your data, and test your predictions against?

2) The biological motivation of the work should be clearer, especially for non-neuroscientists. Explain why these neuronal cell classes are of particular interest. The data are for just four layer-specific but heterogeneous cell classes, and cortical cell types are vastly more complicated than this, but you describe the data as "high resolution". Revise the Introduction and the Discussion to be more clear that these data are of intermediate resolution (albeit at the forefront of a rapidly advancing field) and to explain why inferring a transcriptional network at this level of resolution is biologically useful.

3) If in the Discussion you could make one or two specific, important, experimentally testable predictions from your proposed network, this would be a way of clarifying the biological utility of the data resource, and helping to motivate others to make use of it.

4) Clarify the k-means analysis of modules (subsection “ATAC-seq and RNA-seq module analysis” of Methods). Define precisely what vectors you are clustering, how many dimensions they have, and how you define a distance between them. Explain why you use a two-stage clustering approach. Explain what choosing "cluster centers > 0.5" means: the centroids in a K-means clustering are n-dimensional vectors, not single scalar numbers. Explain what you mean by changing cluster centers from >0.5 to 1: arbitrarily moving k-means centroids makes no sense either. Explain in Figure 5—figure supplement 1 what you mean by "Pearson correlation coefficients for comparisons between cluster centers": two k-means centroids are just two points in n-space, and it's not clear how one calculates a correlation coefficient between two points.

5) Please make the data accessible other than as a raw deposition in GEO. How could an interested biologist use these data with minimal computational overhead? We suggest finding a way to have these data hosted as tracks in a public genome browser such as at UCSC.

6) Clarify whether you believe this TF network is only involved in cell fate maintenance, or its establishment in development, and why.

7) In places, log p-values are being plotted and interpreted as if they measure effect strength (e.g. Figure 4, Figure 5, Figure 6). Be more careful with this. A p-value is a function of both effect size and sample number, and it isn't clear that you are always comparing across equal sample numbers.

---

## [Author Response]

*Essential revisions:*

*1) The lack of experimental validation of any of the predicted roles for transcription factors and their sites is problematic. Are there published data, such as relevant loss-of-function phenotypes, that you can interpret in light of your data, and test your predictions against?*

We have searched the Mouse Genome Informatics (MGI) database for each key regulator to find loss-of-function experiments for each gene. In all but a few cases, previous studies focused on brain regions other than cortex, and examined the role these genes play in development rather than adulthood. We now include discussion of several loss-of-function studies for key transcription factors predicted by our study: *Cux1, Pou3f2*, and *Tbr1*:

“In our analysis, we see largely distinct networks of interactions that define upper and lower layers of the cortex. […] Loss of key TFs can also affect layer-specific cortical projection patterns: knockdown of *Cux1* ablates ipsilateral cortico-cortical projections of upper-layer neurons (Rodriguez-Tornos et al., 2016), and loss of *Tbr1* results in truncated corticothalamic projections by L6 neurons (Hevner et al., 2001).”

While these previous loss-of-function studies are informative, we have also included suggestions that conditional loss-of-function studies be performed to isolate the role that these factors play in specific cell classes (see point 3, below).

In addition, we have expanded our Discussion to include a recent publication which has shown that *Mef2c* is a direct target of FOXP2 through binding of FOXP2 at motifs near *Mef2c* exon 3 in the striatum (Chen et al., Nat. Neurosci., 2016). Mutation of these FOXP motifs in a transient transfection assay demonstrated FOXP2-induced repression. In our network, we also predict that FOXP2 may act as a repressor of *Mef2c* expression. We examined FOXP motif-containing accessible sites near *Mef2c* in our own data, including the previously reported *Mef2c* exon 3-proximal region. The *Mef2c* exon 3-proximal region displayed a modest, but statistically insignificant decrease in chromatin accessibility in lower compared to upper cortical layers in our data. Therefore, it was not included in our analysis. However, we found additional FOXP motif-containing sites upstream of the *Mef2c* TSS that are significantly less accessible in lower vs. upper layers of the cortex, and that we hypothesize contribute to repression of the *Mef2c* gene. It is possible that *Mef2c* repression in stratum and cortex may be achieved through FOXP2- binding of different genomic elements, and further experiments would be needed to address this hypothesis. We address this new study in the Discussion and we include an additional figure (Figure 12).

“A recent publication has shown that FOXP2 directly suppresses *Mef2c* expression in the striatum, where knockout of *Foxp2* allows higher *Mef2c* expression, which, in turn, suppresses synaptogenesis (Chen et al., 2016). […] This hypothesis could be tested by future experiments that measure the chromatin accessibility landscape of the striatum, and genome-wide interaction sites of FOXP2 by ChIP-seq in striatum and cortex, as well as the effect of decreasing FOXP2 levels through cell-type specific knockout or knockdown on these chromatin landscapes.”

*2) The biological motivation of the work should be clearer, especially for non-neuroscientists. Explain why these neuronal cell classes are of particular interest. The data are for just four layer-specific but heterogeneous cell classes, and cortical cell types are vastly more complicated than this, but you describe the data as "high resolution". Revise the Introduction and the Discussion to be more clear that these data are of intermediate resolution (albeit at the forefront of a rapidly advancing field) and to explain why inferring a transcriptional network at this level of resolution is biologically useful.*

We have added an additional paragraph to the Introduction to clarify the reasoning for studying cortical layers:

“Each layer of VISp contains distinct populations of glutamatergic cells with different transcriptional, functional, and connectional properties: layer 4 cells are the primary recipients of the visual signals from the dorsal portion of the lateral geniculate nucleus (LGd); layer 2/3 cells receive signals from L4, and communicate with L5 cells within the same cortical region and other cortical regions; layer 5 cells are highly diverse, and include cells that project to many other cortical and subcortical regions; and the layer 6 cells we examine in this study project to the thalamus (Bortone, 2014; Sorensen, 2015). […] In order to define potential regulatory elements and corresponding transcriptional regulators, we examined chromatin landscapes of these cell classes.”

To more accurately define the resolution of our study, we no longer use “high resolution”:

“[…]our data provide examination of the chromatin accessibility state at the resolution of layer-specific cortical cell classes.”

*3) If in the Discussion you could make one or two specific, important, experimentally testable predictions from your proposed network, this would be a way of clarifying the biological utility of the data resource, and helping to motivate others to make use of it.*

We have added more specific recommendations to the end of the Discussion section:

“Further research into the roles that these and other TFs play in the development and maintenance of layer-specific cell classes will enhance our understanding of the functional consequences of these transcriptomic programs. Conditional loss-of-function experiments in specific cell classes at specific times in development or adulthood would help to disentangle the specific roles from more global roles of these TFs. […] New techniques for ChIP-seq enable enrichment of modified histones from as few as 500 cells (Lara-Astiaso et al., 2014), which may allow the correspondence between accessibility and chromatin modifications in specific cell classes to be studied directly.”

*4) Clarify the k-means analysis of modules (subsection “ATAC-seq and RNA-seq module analysis” of Methods). Define precisely what vectors you are clustering, how many dimensions they have, and how you define a distance between them.*

We now explicitly state the dimensions of the vectors in the Methods section:

“The vectors used for clustering were scaled average peak accessibility values (for peak module analysis) and scaled average gene expression values (for gene module analysis), described above. These vectors each have 4 dimensions, one for each glutamatergic cell class.”

*Explain why you use a two-stage clustering approach. Explain what choosing "cluster centers > 0.5" means: the centroids in a K-means clustering are n-dimensional vectors, not single scalar numbers.*

We performed a two-stage clustering approach because unsupervised k-means clustering is a stochastic process, which randomly selects centroids to establish initial conditions. In order to converge on common modules of peak accessibility and gene expression, we performed an initial round of clustering without providing starting centroids, then used the resulting patterns to seed comparable initial conditions to identify similar modules across both peak accessibility and gene expression datasets. This is now more clearly stated in the Methods section:

“Because unsupervised k-means clustering is dependent on random selection of initial cluster centroids, we selected a set of 8 clusters from the initial k-means cluster results to seed a second round of clustering to find a convergent set of modules.”

*Explain what you mean by changing cluster centers from >0.5 to 1: arbitrarily moving k-means centroids makes no sense either.*

For each of the n-dimensions, we binarized the centroid values to generate 4-dimensional centroid seeds for each cluster in the second round of clustering. This allowed us to use the same starting conditions for both peak and gene clustering. Final centroids of the k-means clusters are not pinned to these values, as the k-means clustering will converge to a result based on multiple iterations (100 in our case). We have changed the wording in the Methods section to reflect this analysis:

“We chose 4 patterns based on selective, high accessibility/expression for each of the glutamatergic cell classes, and 4 other patterns that were frequently observed in both peaks and genes (high in L2/3 and L4, high in L5 and L6, low in L4 only, and low in L6 only). To use these patterns as seed centroids, we generated binarized centroid vectors. For each dimension, values > 0.5 were set to 1, and < 0.5 were changed to 0. We then ran k-means clustering on the selected peaks and genes using these 8 cluster centers as input to assign final peak and gene modules.”

*Explain in Figure 5—figure supplement 1 what you mean by "Pearson correlation coefficients for comparisons between cluster centers": two k-means centroids are just two points in n-space, and it's not clear how one calculates a correlation coefficient between two points.*

In Figure 5—figure supplement 1, we used Pearson correlation to calculate the similarity between the centroid values for each of the initial clusters. We think that the use of Pearson correlation coefficients is reasonable in this limited context, in which we are simply comparing these initial clusters to see how many distinct patterns are common among genes and peaks. The centroid vectors have only 4 dimensions, and we realize that these calculations do not have high statistical power. However, they are still useful for the simple comparisons we perform in Figure 5—figure supplement 1 to assess how similar or dissimilar our initial clusters are.

*5) Please make the data accessible other than as a raw deposition in GEO. How could an interested biologist use these data with minimal computational overhead? We suggest finding a way to have these data hosted as tracks in a public genome browser such as at UCSC.*

We have built a track hub on the UCSC Genome Browser website at: http://genome.ucsc.edu/cgi-bin/hgTracks?db=mm10&hubUrl=http://goldenpathcollaboration:acm0se3@download.alleninstitute.org/goldenpath/hub.txt. We are now working to configure this track hub for inclusion as a registered Public Hub on the UCSC Genome Browser.

*6) Clarify whether you believe this TF network is only involved in cell fate maintenance, or its establishment in development, and why.*

We agree that this is an important, open question in neurodevelopment and epigenetics. While our study provides some insight into the state of TF networks in the adult visual cortex, we feel that extrapolating our results to a role in development is beyond the scope of our data and data available currently in the field. We note in the Discussion that specific interactions previously identified in development or in other cell types are not found in our adult ATAC-seq data (*RORB* regulating *Pou3f2* and POU3F2 regulating *Foxp2*). However, genome-wide data for ATAC-seq in the developing cortex have yet to be generated. To address this question, we have expanded on the possible, but not certain, role that the TF network we uncover in adult cortex may play in development:

“These comparisons emphasize the importance of developmental, anatomical, and cell type context in studies of chromatin accessibility and transcriptomic regulation. […] Future studies performed on layer-specific cell classes in development will be needed to determine how changes to the epigenetic landscape affect transcription during development.”

*7) In places, log p-values are being plotted and interpreted as if they measure effect strength (e.g. Figure 4, Figure 5, Figure 6). Be more careful with this. A p-value is a function of both effect size and sample number, and it isn't clear that you are always comparing across equal sample numbers.*

We agree with this reviewer, and we have been careful throughout the manuscript with p-value interpretation to avoid statements that are misleading. In our interpretation of the results presented in Figure 4, Figure 5, and 6, we treat these p-values as independent measures of significance, and do not use them to compare between calculations with different sample sizes. The p-values are also not used to define association criteria for building the TF network, as outlined in Figure 7—figure supplement 1.

To assist readers in interpreting these results, we now include a supplementary figure that shows the log-odds ratios for Figure 5 so that effect size and p-value are both present. Both p-value and log-odds are also provided in the figure source data file we have provided for Figure 5 ([Supplementary-material SD11-data]). For Figure 4, we do not present an odds ratio, because this is a non-parametric test of significance (Mann-Whitney). For Figure 6, adjusted p-values were calculated using AME, which does not output an odds ratio for Fisher’s exact test calculations.